# NeuroGraph: Benchmarks for Graph Machine Learning in Brain Connectomics

**Anwar Said**
Vanderbilt University
anwar.said@vanderbilt.edu

**Roza G. Bayrak**
Vanderbilt University
roza.g.bayrak@vanderbilt.edu

**Tyler Derr**
Vanderbilt University
tyler.derr@vanderbilt.edu

**Mudassir Shabbir**
Vanderbilt University
Information Technology University
mudassir.shabbir@itu.edu.pk

**Daniel Moyer**
Vanderbilt University
daniel.moyer@vanderbilt.edu

**Catie Chang**
Vanderbilt University
catie.chang@vanderbilt.edu

**Xenofon Koutsoukos**
Vanderbilt University
xenofon.koutsoukos@vanderbilt.edu

## Abstract

Machine learning provides a valuable tool for analyzing high-dimensional functional neuroimaging data, and is proving effective in predicting various neurological conditions, psychiatric disorders, and cognitive patterns. In functional magnetic resonance imaging (MRI) research, interactions between brain regions are commonly modeled using graph-based representations. The potency of graph machine learning methods has been established across myriad domains, marking a transformative step in data interpretation and predictive modeling. Yet, despite their promise, the transposition of these techniques to the neuroimaging domain has been challenging due to the expansive number of potential preprocessing pipelines and the large parameter search space for graph-based dataset construction. In this paper, we introduce NeuroGraph[1], a collection of graph-based neuroimaging datasets, and demonstrated its utility for predicting multiple categories of behavioral and cognitive traits. We delve deeply into the dataset generation search space by crafting 35 datasets that encompass static and dynamic brain connectivity, running in excess of 15 baseline methods for benchmarking. Additionally, we provide generic frameworks for learning on both static and dynamic graphs. Our extensive experiments lead to several key observations. Notably, using correlation vectors as node features, incorporating larger number of regions of interest, and employing sparser graphs lead to improved performance. To foster further advancements in graph-based data driven neuroimaging analysis, we offer a comprehensive open-source Python package that includes the benchmark datasets, baseline implementations, model training, and standard evaluation.

---

[1]https://anwar-said.github.io/anwarsaid/neurograph.html

37th Conference on Neural Information Processing Systems (NeurIPS 2023) Track on Datasets and Benchmarks.

Table 1: Dataset statistics. $|G|$ denotes the number of graphs, $|N|_{avg}$ and $|E|_{avg}$ denote the average number of nodes and edges, $d$ indicates the degree, and $K$ signifies the global clustering coefficient.

| | Dataset | Statistics | | | | | | Node Feat. (dim) | #Classes | Prediction Task |
|---|---|---|---|---|---|---|---|---|---|---|
| | | $|G|$ | $|N|_{avg}$ | $|E|_{avg}$ | $d_{max}$ | $d_{avg}$ | $K_{avg}$ | | | |
| **Static** | HCP-Task | 7443 | 400 | 7029.18 | 153 | 17.572 | 0.410 | 400 | 7 | Graph Classification |
| | HCP-Gender | 1078 | 1000 | 45578.61 | 413 | 45.579 | 0.466 | 1000 | 2 | Graph Classification |
| | HCP-Age | 1065 | 1000 | 45588.40 | 413 | 45.588 | 0.466 | 1000 | 3 | Graph Classification |
| | HCP-FI | 1071 | 1000 | 45573.67 | 413 | 45.574 | 0.466 | 1000 | - | Graph Regression |
| | HCP-WM | 1078 | 1000 | 45578.61 | 413 | 45.579 | 0.466 | 1000 | - | Graph Regression |
| **Dynamic** | DynHCP-Task | 7443 | 100 | 843.04 | 57 | 8.430 | 0.427 | 100 | 7 | Graph Classification |
| | DynHCP-Gender | 1080 | 100 | 874.88 | 53 | 8.749 | 0.439 | 100 | 2 | Graph Classification |
| | DynHCP-Age | 1067 | 100 | 875.42 | 53 | 8.754 | 0.439 | 100 | 3 | Graph Classification |
| | DynHCP-FI | 1073 | 100 | 874.82 | 53 | 8.748 | 0.438 | 100 | - | Graph Regression |
| | DynHCP-WM | 1080 | 100 | 874.88 | 53 | 8.749 | 0.439 | 100 | - | Graph Regression |

# 1 Introduction

Graph Neural Networks (GNNs) have demonstrated remarkable efficacy in a variety of domains including recommendations, forecasting, and biomedical data analysis [1, 2, 3]. In human neuroimaging research, GNNs have proven valuable in capturing the complex connectivity patterns within brain functional networks but also enhance the modeling and analysis with other relevant informative features [4, 3]. For instance, examining synchronized fluctuations of functional magnetic resonance imaging (fMRI) signals provides a useful means of measuring functional network connectivity [5].

Neuroimaging and Graph Machine Learning (GML) are two rapidly evolving fields with immense potential for mutual collaboration. However, a significant challenge lies in bridging the gap between these domains and enabling seamless integration of neuroimaging data into state-of-the-art GML approaches [6]. This gap is primarily attributed to the expansive number of potential fMRI data preprocessing workflows, the absence of an intuitive tool to generate fMRI graph representations for graph-based learning approaches, and the knowledge gap between the fields of neuroimaging and advanced graph machine learning [2]. To address these challenges, the main objectives of this study are, first, a careful exploration of graph-based dataset generation with the goal of formulating a roadmap for graph-based representations of fMRI data. Second, we conduct a rigorous evaluation of graph machine learning methodologies, with a special emphasis on GNNs, examining their efficacy when applied to diverse fMRI data configurations.

The human brain, a complex network of interconnected regions, can be represented as a graph, wherein nodes correspond to contiguous segments known as Regions of Interest (ROIs), and edges represent their relationships [7, 8]. Features of the functional connectome, such as correlations between the BOLD (Blood Oxygen Level Dependent) signals between different brain regions, typically employed for downstream machine learning tasks [6, 9], can be re-envisioned as node features within attributed graph representations. These representations pave the way for a rich assortment of graph-based data representations, wherein GNNs are exceptionally well-suited [10]. Yet, the vast potential offered by the intersection of fMRI datasets and GNNs remains untapped, due primarily to the expansive search space for data generation and the multifaceted nature of hyperparameters. In this study, we pioneer a rigorous exploration and benchmarking for GNNs, with the following primary contributions:

- We introduce NeuroGraph, a collection of static and dynamic brain connectome datasets tailored for benchmarking GNNs in classification and regression tasks including gender, age, task classification, and prediction of fluid intelligence and working memory scores. This enables an extensive exploration of brain connectivity and its associations with various cognitive, behavioral, and demographic variables. Details of the proposed datasets are provided in Table 1.

- We perform an extensive exploratory study in search of optimal graph-based data representations for neuroimaging data, implementing 15 baseline models on 35 different datasets. Additionally, we provide detailed benchmarking for the datasets we propose.

By offering NeuroGraph, we create an essential bridge between the neuroimaging and graph machine learning communities. Researchers in the neuroimaging field can more readily tap into the power of cutting-edge GNNs. Specifically, our datasets generation pipeline may guide researchers toward effectively transforming neuroimaging data into a unified graph representation suitable for graph machine learning. This integration facilitates the adoption of state-of-the-art graph-based techniques, unlocking new insights and accelerating discoveries in the neuroimaging field.

## 2 Related Work

While functional brain connectomes have long been recognized as a rich source of information in neuroscience and neuroinformatics [11, 12], their value has become increasingly evident in recent years [13]. Propelled by growth in data availability and methodological breakthroughs, ML has shown remarkable efficacy on tasks such as decoding of cognitive processes [14, 15] and diagnosing mental health disorders [16, 17]. Simultaneously, there has been increased utilization of static and dynamic graph representation learning methods for brain analytics, which we briefly summarize in this section.

**Static graph representations:** GNNs have significantly evolved as a major field of exploration, offering an intuitive approach in learning from graph-structured data [18, 19, 20, 21, 22]. In a static setting, where individual data points are represented by single graphs, a variety of methods have been introduced [23, 21, 24, 22, 25, 26]. Recent studies have demonstrated the effectiveness of various approaches when applied to functional connectome data, which can be represented as different types of graphs, including weighted graphs [3, 27, 28], and attributed graphs [2, 29], among others. In benchmarking setup, BrainGB [27] stands out as a notable advancement, offering a unified framework for brain network analysis utilizing GNNs. Likewise, BrainGNN [3] introduces a specialized GNN architecture tailored for the discovery of neurological biomarkers from fMRI data. By leveraging the structured nature of the data and incorporating local information, GNNs not only enable learning from the functional connectivity patterns but also enhance modeling and analysis with other relevant informative features [9, 30, 22, 1, 31].

**Dynamic graph representations:** The field of learning dynamic graph representations in a graph machine learning setting remains relatively unexplored, especially in the realm of brain imaging [32]. In neuroimaging, dynamic graphs are constructed to capture the time-varying interactions and connectivity patterns in the brain [2, 33, 34]. While GML methods have not been commonly employed in this domain, recent years have seen the introduction of methods that yield impressive results on dynamic brain graphs[2, 35, 36, 37, 38]. Specifically, Kim et al. [2] have made significant strides by introducing dynamic GNNs tailored specifically for brain graphs. These methods have showcased the potential of effectively capturing and analyzing the dynamic nature of brain connectivity, opening up new avenues for advancements in our understanding of brain function and neurological processes.

## 3 NeuroGraph

The design space for graph construction from functional connectome is vast, since a variety of methods can be employed to generate various forms of graphs, such as simple undirected graphs [2], weighted graphs[3], attributed graphs [27], and minimum spanning trees [39, 40], among others. Thoroughly navigating this extensive design space and evaluating all potential parameter combinations is a challenging task. While recent efforts have been undertaken to leverage GNNs for predictive modeling on neuroimaging data, a consensus has yet to be reached regarding the preprocessing pipeline and hyper-parameter configurations best suited for generating expressive graph-based neuroimaging datasets [2, 3, 41, 28, 42]. In addition, although there are a multitude of GNNs models, no benchmark datasets have been created to evaluate GML approaches on brain connectome data. To fill this gap and provide a common ground between neuroimaging and GML communities, we use publicly available datasets and only minimally preprocess the data using standard fMRI preprocessing steps. We provide an illustration of the overall NeuroGraph preprocessing pipeline in Figure 1.

### 3.1 From fMRI to Graph Representations

fMRI data is typically represented in four dimensions, where the blood-oxygen level-dependent (BOLD) signal is captured over time in a series of 3-dimensional volumes. These volumes display the intensity of the BOLD signal for different spatial locations in the brain. However, since brain activity tends to exhibit strong spatial correlations, the BOLD signal is often summarized into a collection of special functional units, *brain parcels*. These units represent *regions of interest* (ROIs) whose constituent "voxels" (a smallest three dimensional resolution) exhibit temporally correlated activity.

The Human Connectome Project (HCP) [43] is a publicly available rich neuroimaging dataset containing not only imaging data but also a battery of behavioral and cognitive data. We select this dataset for benchmarking and utilize the established group level Schaefer [44] atlases to represent the

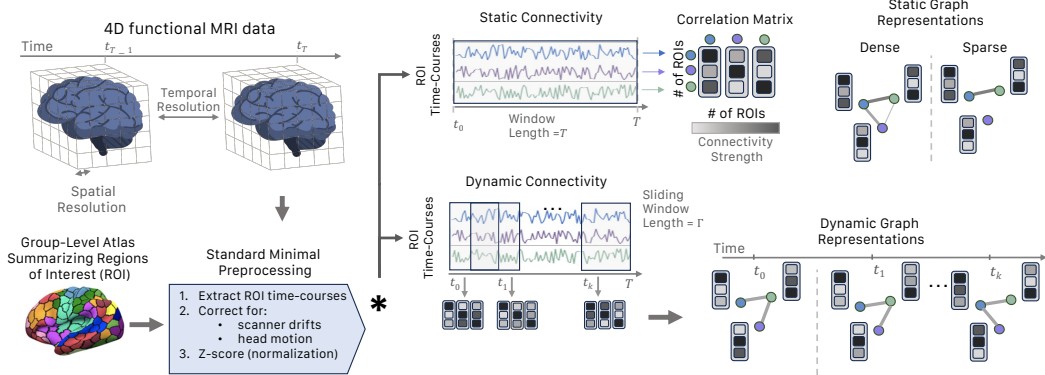

Figure 1: An illustration of the preprocessing pipeline, demonstrating the transition from fMRI data to the construction of both static and dynamic graphs.

measured BOLD signal. These atlases provide a parcellation of the cerebral cortex into hierarchically organized regions at multiple granularities (resolutions).

We use resting-state and seven task fMRI paradigms from the HCP 1200 dataset. All fMRI scans underwent the HCP minimal preprocessing pipeline [45]. We further regressed out six rigid-body head motion parameters and their derivatives, as well as the low-order trends, from the minimally preprocessed data. The mean fMRI time series was extracted from all voxels within each ROI for different parcellation schemes. Individual (subject-wise) ROI time-series signals were temporally normalized to zero mean and unit variance. In summary, our proposed fMRI preprocessing pipeline encompasses five steps: a) brain parcellation, removal of b) scanner drifts and motion artifacts, c) subject-level signal normalization, d) calculation of correlation matrices and finally, e) construction of static and dynamic brain graphs. For further in-depth details, we refer the reader to Appendix B.3.

Our study of these datasets encompasses two distinct modes of analysis: *static* and *dynamic* graph construction. We apply different GNNs to both types and perform benchmarking in five unique tasks. In the static graph construction, we investigate multiple parameters to build the graphs from the raw data, taking into consideration variations in node features, the number of nodes or regions of interest (ROIs), and the density of the graph. For node features, we take into account correlations, time-series signals, or a blend of both. For the number of nodes provided by [44] (i.e., ROIs), we examine three different resolutions: 100, 400, and 1000 nodes. As for graph density, we consider sparse, medium, and dense configurations. For the sparse setup, we choose the top 5% of values from the correlation matrix for edge selection, whereas for the medium and dense setups, we select the top 10% and 20% of values, respectively. We note that constructing brain graphs involves a number of parameters including the choice of parcellation methods, where various brain atlases can be employed to segment the brain into regions of interest. Equally crucial is defining the number of ROIs, as it wields a significant influence on exploring brain functions. The selection of ROIs count permits the creation of brain graphs of varying sizes, allowing the opportunity to focus on both the global and granular levels of the brain. We have opted for those more likely to yield superior performance [3, 2]. Additional details about the complexity of the search space in benchmark dataset construction and the rationale behind these parameters are presented in Appendix B.4. We performed extensive experiments in the exploratory analysis to find the suitable combination of parameters and use a total of 15 baseline methods for benchmarking. The baselines include 10 GNNs, 3 conventional machine learning methods and 2 new architectures. Using the optimal combination of parameters in the static setting, we generate benchmark datasets for corresponding tasks in the dynamic setting. In the subsequent sections, we first describe the generation of graph-based datasets, followed by the description of each task.

## 3.2 Graph Representation

The landscape of constructing brain graphs encompasses a variety of approaches. Previous research in this domain has explored various methodologies for constructing brain graphs and subsequent downstream tasks. For instance, in [2, 3, 46, 47], distinct measures such as mean activation, node index as coordinates, spatial one-hot encoding, and correlations have been employed as node features

with brain graphs. Additionally, a diverse range of measures including partial correlation, Pearson correlation, and geometric distances, among others, have found wide application in defining edges within brain graphs [48]. Our static graph representation encompasses the conventional methodology of generating a static functional connectome graph from an fMRI scan, see Appendix, B.3 for additional details. We define a connectome graph as $G = (\mathcal{V}, \mathcal{E}, X)$, wherein the node set $\mathcal{V} = \{v_1, v_2, \ldots, v_n\}$ represents ROIs, while the edge set $\mathcal{E} \subseteq \mathcal{V} \times \mathcal{V}$ represents positive correlations between pairs of ROIs, determined via a defined threshold. The feature matrix is denoted by $X \in \mathbb{R}^{n \times d}$, where $n$ signifies the total number of ROIs and $d$ refers to the feature vector's dimension. In our benchmarking setup, we define correlation vectors as node features. Subsequently, we define a representation vector $\boldsymbol{h}_G$ for the graph $G$, obtained via a GNN with an objective to perform the desired downstream machine learning task.

fMRI data comprise numerous timepoints within a scan, permitting the construction of dynamic graphs and thereby emphasizing the temporal information encapsulated within the data. This strategy has been evidenced to be notably effective within the literature [2]. Within the dynamic context, we define a sequence of brain graphs over $T$ timepoints, denoted as $\mathcal{G} = \{G_1, G_2, \ldots G_T\}$, wherein each graph $G_t$ captured at index $t$ to $t + \Gamma$ from the fMRI scan. Here, $\Gamma$ signifies the window length, set to 50 with a stride of 3 in our experiments. This setup allows us to capture functional connectivity within 36 seconds every 2.16 seconds, adhering to a common protocol for sliding-window analyses as outlined in [34]. Following the approach in [2], we opt to randomly crop the ROI-timeseries data to a length of 150 timepoints. This procedure results in a total of 34 frames per subject, mitigating the computational and memory overhead in training complex models.

The procedure for constructing a graph at each time-point similar to the one applied to the static graph. The initial preprocessing, including parcellation, noise removal, and addressing head motions, remains consistent in order to construct the timeseries object. For each window, individual normalization has been performed, and then correlation matrices and corresponding graphs are constructed. Subsequently, $\mathcal{G}$ can be utilized to generate a dynamic graph representation $h_{dyn}$ to execute the desired downstream ML task. We refer the reader to the Appendix B.3 for further details.

## 3.3 Benchmark Datasets

The benchmark datasets are primarily divided into three main categories: those constructed for classification of demographics and task states, and those constructed for estimation of cognitive traits. Each category encapsulates distinct aspects of the collected data and serves unique analytical purposes. A brief description is provided below for each of these categories with some basic statistics presented in Table 1.

**Predicting Demographics:** The category of demographic estimation in our dataset is comprised of gender and age estimation [41]. The gender attribute facilitates a binary classification with the categories being male and female. Age is categorized into three distinct groups as in [27]: 22-25, 26-30, and 31-35 years. A fourth category for ages 36 and above was eliminated as it contained only 14 subjects (0.09%), to maintain a reasonably balanced dataset. We introduce four datasets named: HCP-Gender, HCP-Age, DynHCP-Gender, and DynHCP-Age under this category. The first two are static graph datasets while the last two are the corresponding dynamic graph datasets.

**Predicting Task States:** The task-decoding involves seven tasks: Emotion Processing, Gambling, Language, Motor, Relational Processing, Social Cognition, and Working Memory. Each task is designed to help delineate a core set of functions relevant to different facets of the relation between human brain, cognition and behavior [49]. Under this category, we present two datasets: HCP-Task, a static representation, and DynHCP-Task, its dynamic counterpart.

**Estimating Cognitive Traits:** The cognitive traits category of our dataset comprises two significant traits: working memory (evaluated with List Sorting) [50] and fluid intelligence (evaluated with PMAT24) [51]. Working memory refers to an individual's capacity to temporarily hold and manipulate information, a crucial aspect that influences higher cognitive functions such as reasoning, comprehension, and learning [52]. Fluid intelligence represents the ability to solve novel problems, independent of any knowledge from the past. It demonstrates the capacity to analyze complex relationships, identify patterns, and derive solutions in dynamic situations [20, 30]. The prediction of both these traits, quantified as continuous variables in our dataset, are treated as regression problem. We aim to predict the performance or scores related to these cognitive traits based on the func-

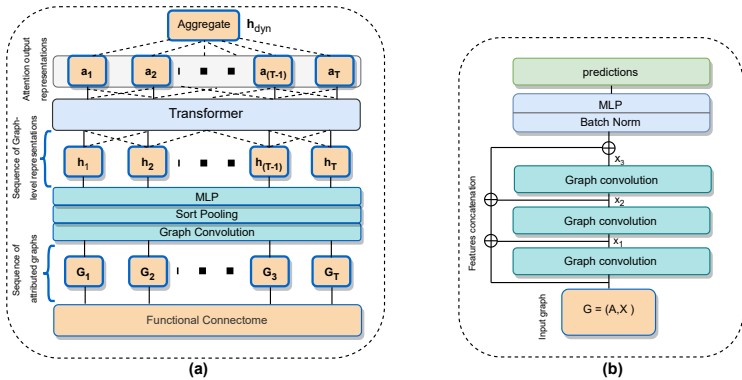

Figure 2: (a). Illustration of the architecture for learning dynamic graph representations. (b). Visualization of the GNN* architecture featuring residual connections and concatenated features.

tional connectome graphs. We generate four datasets under cognitive traits: HCP Fluid Intelligence (HCP-FI), HCP Working Memory (HCP-WM), DynHCP-FI and DynHCP-WM.

## 3.4 Learning models

The functional connectome, which effectively captures the network structure of brain activity, has proven to be a valuable representation of fMRI data for machine learning, as demonstrated in numerous previous studies and our own experiments [4, 9]. Recognizing its significance in the learning process, we sought a suitable GNN framework that could effectively leverage the comprehensive functional connectome data through a combination of message passing and neural network. After thorough exploration, we implemented a GNN architecture, denoted as GNN* illustrated in Figure 2 (b), that incorporates residual connections and concatenates hidden representations obtained from message passing at each layer. To further enhance the model's performance, we employ batch normalization and a multi-layer perceptron (MLP) to effectively utilize the combined representations during training. While adaptive residual connections have been extensively explored in GNNs, we present this simple and unique architecture for brain graphs that effectively learns the representations for brain graphs [53].

Recently, a number of dynamic graph representation approaches in conjunction with recurrent neural networks (RNNs) such as GRU, LSTM, and transformers, have been introduced [33, 54]. However, assessing the effectiveness of GNN models in a unified dynamic setting using the existing approaches presents a significant challenge. Therefore, we implement a simple and generalized architecture tailored to process dynamic graphs for the graph classification problem, as illustrated in Figure 2 (a). Our architecture comprises two distinct modules. The first is a GNN-based learning module, responsible for deriving graph-level representations from each of the derived graph snapshot. Following this, a transformer module takes over, applying attention to the learned representations from the GNNs. Finally, the outputs are averaged into a single dynamic graph representation vector, $h_{dyn}$. This design offers a universally applicable method for evaluating multiple GNN methods within a dynamic graph setting for the downstream ML classification and regression problem.

## 4 Benchmarking Setup

In order to thoroughly evaluate the performance of brain graphs generated through different hyperparameters, we propose a series of questions, defined as hyperparameter probe. These questions seek to identify the optimal hyperparameter setting for our graph-based neuroimaging analysis and ultimately enhance the performance of the predictive models derived from it.

**Question 4.1** *What are the optimal node feature configurations?*

The first question aims to identify the best configurations for node features. This involves an exploration and comparison of various feature representations to discern their effectiveness on the performance of the derived predictive models. In assessing node feature configurations, our analysis encompasses the correlation matrix, the time-series BOLD signals, as well as their combination. The

correlation matrix is generated by calculating the correlation values amongst all ROIs. On the other hand, the BOLD signals are derived post the preprocessing of the input fMRI image, adhering to the preprocessing pipeline outlined in Section 3.1.

**Question 4.2** *To what extent does the number of ROIs impact the performance of predictive modeling on graphs?*

The second question delves into the influence of varying the number of ROIs on the performance of predictive modeling. The objective is to assess how the granularity of ROIs affects the quality and the performance of the predictive models. We evaluate the use of $100, 400$ and $1000$ ROIs.

**Question 4.3** *To what degree does sparsifying brain functional connectome graphs impact the performance of predictive modeling? What threshold yields optimal performance?*

Our third question investigates the impact of sparsifying brain functional connectome graphs on the performance of the predictive models. It aims to establish a threshold that leads to optimal model performance in graph machine learning setting. In our exploration, we consider the top 20%, 10%, and 5% percentile values from the correlation matrices to construct the graph edges.

**Question 4.4** *Which graph convolution approaches are preferable for the predictive modeling on brain graphs?*

Our fourth and final question delves into the exploration of various graph convolution methods, assessing their suitability for predictive modeling on brain graphs. The aim here is not only to identify, but also to recommend the most effective techniques, considering the specific features and intricacies of neuroimaging data. In this endeavor, we have put to test over 12 GNNs, which include two of our own implemented frameworks, to gauge their comparative performance.

By addressing these questions, we aim to set a robust benchmarking framework for graph-based machine learning methods in neuroimaging, providing invaluable insights into their optimal application.

# 5 Benchmarking Results

In this section, we introduce the baseline models, describe our experimental setup, and present the results from our preliminary exploration study. Following this, we lay out our approach to benchmarking.

## 5.1 Baselines and Experimental Setup

This section outlines the specifics of our unique, generalized experimental setup designed to evaluate a range of GNN models. We consider 10 well-established GNN models: $k-$GNN [22], GCN [18], GraphSAGE [19], Unified Message Passing (UniMP)[25], Residual GCN (ResGCN) [21], Graph Isomorphism Network (GIN), Chebyshev Convolution (Cheb) [24], Graph Attention Network (GAT) [23], Simplified GCN (SGC) [55], and General Convolution (General) [56][2]. We also consider 3-layered Neural Network (NN), 2D Convolutional Neural Network (CNN) and Random Forest (RF) for the comparison.

In our experimental setup, we devise a graph classification architecture comprising three layers of GNNs, followed by a sort pooling aggregator [57]. Sort pooling sorts the node features based on the last channel, selecting only the first $k$ representations. Subsequently, sort pooling is advanced through two one-dimensional convolution layers, which are then succeeded by a two-layer Multi-Layer Perceptron (MLP). This architecture has been consistently utilized across all GNNs throughout the entire experimental setup. For the dynamic datasets, we utilize our baseline method with five different GNNs. For NN, we utilized 512, 256, and 128 hidden units in each layer, respectively. For the CNN, we utilized a four-layer model with a stride of 2, 64 kernels of size 5, and padding set to 2. This was complemented by three fully connected layers [58]. For the Random Forest (RF) [59], we opted for 100 estimators, leaving the remaining parameters at their Scikit-learn defaults. All of our experiments were carried out on a system equipped with an Intel(R) Xeon(R) Gold 6238R CPU operating at 2.20GHz with 112 cores, 512 GB of RAM, and an NVIDIA A40 GPU with 48GB of memory.

---

[2]we use PyG implementations and default settings for running all these models

Table 2: Results of the gender classification using three distinct node feature configurations across three settings, evaluated on 10 GNNs. The configurations include CORRELATIONS (CORR), BOLD signals, and a combination of BOLD + CORRELATIONS, evaluated across 100ROIs, 400ROIs, and 1000ROIs. Avg. column indicates the average results across the row and numbers under ROIs indicates average results across each ROI. The blue notation highlights the overall best results for each GNN and red indicates average best performance across each ROI. Average best results are obtained through 1000 ROIs with sparser graphs.

| Dataset | | k-GNN | GCN | SAGE | UniMP | ResGCN | GIN | Cheb | GAT | SGC | General | Avg. |
|---|---|---|---|---|---|---|---|---|---|---|---|---|
| 100ROIs | CORR | 65.65 | 68.98 | 68.70 | 68.33 | 66.06 | 68.24 | 63.94 | 69.49 | 68.43 | 64.95 | **67.30** |
| | BOLD | 49.58 | 50.97 | 51.67 | 51.30 | 51.34 | 55.09 | 53.19 | 49.95 | 51.90 | 51.11 | 51.11 |
| | CORR+BOLD | 52.78 | 51.02 | 50.28 | 50.79 | 50.60 | 54.91 | 49.44 | 50.37 | 51.57 | 51.30 | 51.36 |
| 400ROIs | CORR | 72.21 | 74.10 | 61.66 | 68.57 | 70.09 | 71.89 | 58.94 | 69.35 | **75.99** | **73.09** | **69.56** |
| | BOLD | 51.16 | 51.62 | 53.94 | 51.39 | 52.31 | 55.09 | 49.07 | 50.46 | 53.24 | 53.94 | 52.22 |
| | CORR+BOLD | 51.53 | 51.90 | 52.96 | 51.57 | 52.36 | 55.56 | 50.63 | 52.13 | 52.08 | 52.61 | 53.33 |
| 1000ROIs | CORR | **78.80** | **75.19** | **71.71** | **75.14** | **78.75** | **77.22** | **64.77** | **71.34** | 73.75 | 63.13 | **72.98** |
| | BOLD | 48.15 | 46.99 | 49.31 | 50.93 | 47.92 | 56.48 | 47.22 | 50.93 | 49.31 | 51.62 | 49.89 |
| | CORR+BOLD | 51.30 | 51.81 | 51.25 | 51.11 | 49.86 | 54.35 | 49.66 | 51.22 | 51.34 | 51.37 | 51.33 |

Models training: We have carefully carried out the training and evaluation of each dataset in our study. Each dataset was partitioned randomly with 70% training, 20% testing, and 10% for validation. To ensure reproducibility and balance across the datasets, we employed a fixed seed, 123, for the split in a stratified setting. This stratified approach facilitated an equitable distribution of classes in each partition. Each model underwent training for 100 epochs with a learning rate of $1e^{-5}$ for classification, and for 50 epochs with a learning rate of $1e^{-3}$ for regression problem. Across all experiments, we set dropout to 0.5, weight decay to $5e^-4$, and designated 64 hidden dimensions for both the GNN convolution and MLP layers. Furthermore, for loss functions, we utilized cross entropy for classification and mean absolute error for regression problems. All benchmarks and their source codes can be accessed on GitHub[3]. The static benchmark datasets are also available at PyG[4].

## 5.2 Exploratory Experiments and Results

Here we address the questions outlined earlier by conducting a series of experiments including the evaluation of different node feature configurations, the influence of varying numbers of ROIs, the implications of sparsity in brain graphs, and the effectiveness of diverse graph convolution approaches. Each experiment aligns with a question, thereby paving the way for comprehensive analysis and definitive conclusions.

**Performance enhancement with correlations as node features:** Our first step involves evaluating the interplay between the number of ROIs and the configuration of node features, with an aim to streamline the overall search space. For this purpose, we engage in the gender classification problem using 10 different GNNs. The results of these experiments are presented in Table 2. It is clear that employing correlations as node features consistently enhances the performance across all evaluated numbers of ROIs. However, what caught our attention was the significant variance in the results obtained through correlations and BOLD signals and the number of ROIs. The performance notably declines when correlations and BOLD signals are combined, and the number of ROIs are reduced. This motivates further investigation on how to leverage BOLD signal or perhaps obtain features from the BOLD signals to be used for learning. Furthermore, the performance of different GNNs baselines does not consistently correlate with the number of ROIs or node features.

**Performance enhancement through large ROIs and sparse brain graphs:** Our analysis extended to evaluating the efficacy of 10 GNNs on gender classification, using a varying number of ROIs and different graph densities. In addition to gender classification, we further incorporated task-state classification problem to strengthen our observations under different settings. For all the experiments, we opted for correlations as node features, a decision driven by the consistent boost they offer in performance from the last experiment. The results are presented in Table 3. An important observation from our findings reveals that larger numbers of ROIs, (1000) demonstrate superior performance in gender classification. Similarly, a significant number of GNNs exhibit improved results with the use of 1000 ROIs for the task-state classification problem. An analysis of the graph densities reveals an

---

[3]https://github.com/Anwar-Said/NeuroGraph
[4]https://pytorch-geometric.readthedocs.io/

Table 3: Performance comparison in terms of accuracy of 10 GNNs with different ROIs and varying graph densities on gender and task state classification problems. The blue highlights the overall best results per model in each classification problem.

| Dataset | | | $k-$GNN | GCN | SAGE | UniMP | ResGCN | GIN | Cheb | GAT | SGC | General |
|---|---|---|---|---|---|---|---|---|---|---|---|---|
| Gender Classification | 100ROIs | Sparse | 63.33 | 72.96 | 69.35 | 69.72 | 68.06 | 69.72 | 63.70 | 70.28 | 70.37 | 67.22 |
| | | Medium | 65.65 | 68.98 | 68.70 | 68.33 | 66.06 | 68.24 | 63.94 | 69.49 | 68.43 | 64.95 |
| | | Dense | 64.44 | 68.52 | 65.00 | 68.06 | 66.39 | 63.70 | 64.26 | 69.72 | 68.43 | 61.76 |
| | 400ROIs | Sparse | 69.95 | **77.14** | 69.86 | 67.56 | 71.43 | 69.4 | 66.45 | 72.72 | **78.25** | 76.13 |
| | | Medium | 65.65 | 68.98 | 68.70 | 68.33 | 66.06 | 68.24 | 63.94 | 69.49 | 68.43 | 64.95 |
| | | Dense | 71.61 | 76.13 | 62.58 | 61.20 | 69.77 | 73.27 | 61.84 | 67.83 | 74.19 | 72.44 |
| | 1000ROIs | Sparse | **82.13** | 75.46 | 77.69 | **76.67** | 78.33 | 75.56 | 59.07 | **76.2** | 76.48 | **78.89** |
| | | Medium | 78.80 | 75.19 | 71.71 | 75.14 | 78.75 | 77.22 | 71.43 | 71.34 | 73.75 | 63.13 |
| | | Dense | 61.57 | 73.80 | **78.86** | 72.50 | **78.89** | **78.70** | **76.67** | 71.67 | 75.25 | 72.69 |
| Task Classification | 100ROIs | Sparse | 91.50 | 91.56 | 91.43 | 92.73 | 92.14 | 88.31 | 92.55 | 92.91 | 91.40 | 91.52 |
| | | Medium | 90.91 | 90.80 | 91.81 | 92.75 | 92.25 | 88.01 | 93.06 | 93.15 | 91.40 | 91.22 |
| | | Dense | 90.30 | 91.15 | 93.15 | 93.28 | 93.02 | 87.12 | 93.18 | 93.08 | 90.49 | 89.47 |
| | 400ROIs | Sparse | 93.23 | **94.21** | 94.78 | 94.72 | 94.61 | **89.79** | 94.45 | 95.2 | **94.17** | 93.62 |
| | | Medium | 92.26 | 93.93 | 93.89 | 95.02 | 94.33 | 89.44 | 79.03 | 94.67 | 93.39 | 93.58 |
| | | Dense | 90.64 | 93.36 | **95.76** | 94.48 | **94.64** | 88.22 | 87.24 | 94.78 | 93.18 | 90.84 |
| | 1000ROIs | Sparse | 93.50 | 93.80 | 94.09 | 93.59 | 94.23 | 85.14 | 93.82 | 94.66 | 93.2 | **94.17** |
| | | Medium | 92.65 | 90.87 | 94.39 | **95.79** | 92.04 | 85.40 | **94.88** | 94.00 | 91.37 | 91.87 |
| | | Dense | **93.77** | 93.12 | 94.12 | 94.54 | 93.59 | 81.59 | 92.92 | **95.35** | 93.76 | 93.76 |

Table 4: Classification results in terms of accuracy on benchmark static datasets constructed with optimal setting. Blue indicates overall best results.

| Dataset | NN | CNN | RF | $k-$GNN | GCN | SAGE | UniMP | ResGCN | GIN | Cheb | GAT | SGC | General | GNN* |
|---|---|---|---|---|---|---|---|---|---|---|---|---|---|---|
| HCP-Task | 97.78 | 95.88 | 88.98 | 93.23 | 94.21 | 94.78 | 94.72 | 94.61 | 89.79 | 94.45 | 95.2 | 94.17 | 93.62 | **98.20** |
| HCP-Gender | 86.67 | 76.39 | 69.9 | 82.13 | 75.46 | 77.69 | 76.67 | 78.33 | 75.56 | 59.07 | 76.20 | 76.48 | 78.89 | **89.07** |
| HCP-Age | 44.23 | 43.38 | 40.84 | 42.72 | 43.66 | 40.94 | 43.85 | 40.00 | 44.98 | 41.97 | 42.25 | 43.47 | 41.03 | **50.23** |

intriguing trend. For instance, based on the results from Table 3, the ratio of sparse:medium:dense on the gender classification dataset is $6 : 0 : 4$, while on the task dataset, it stands at $4 : 2 : 4$ for 1000 number of ROIs. Furthermore, the differences in results, especially in cases where sparse graphs exhibit lower performance, are generally small. Recognizing the increased complexities stemming from memory usage, training demands, and the possibility of oversmoothing, we have chosen sparse graphs with the combination of large ROIs, and correlation features in our benchmarking setup.

### 5.3 Benchmarking with Optimal Settings

Considering the optimal setting obtained through exploring search space presented in the previous section, here we present the experimental setup and benchmarking results on the proposed 10 datasets.

The classification accuracy of all baseline models is detailed in Table 4. It is evident from the results that the GNN* stands out as the leading performer. However, the Neural Network's performance is also notably impressive. Similarly, the results pertaining to the regression problems have been outlined in Table 5. The leading performer on the regression problems is again GNN*. These results distinctly demonstrate that *residual connections* coupled with message passing play a pivotal role in enhancing performance in brain networks. This synergy arises from the capacity of message passing to glean meaningful representations from highly correlated features. Simultaneously, the inclusion of residual connections empowers the utilization of the input features with the learned representations obtained through message passing. This also underscores the effectiveness of correlation features in influencing the model's performance on brain graphs.

In Table 6, we lay out the classification and regression results obtained on the dynamic datasets. Given the consideration of a basic dynamic baseline and the construction of dynamic datasets using limited dynamic lengths and number of ROIs, the performance does not quite match up to the static datasets. Nonetheless, it's worth noting that UniMP, despite the constraints, consistently demonstrates competitive performance.

## 6 Conclusion and Future Works

In this work, we introduce novel brain connectome benchmark datasets specifically tailored for graph machine learning, representing a promising avenue for addressing various challenges in neuroimaging. The inherent symmetries and complex higher-level patterns found in brain graphs make them well-

Table 5: Results for HCP-FI and HCP-WM dataset using mean absolute error (MAE). Blue indicates overall best results.

| Dataset | $k$-GNN | GCN | SAGE | UniMP | ResGCN | GIN | Cheb | GAT | SGC | General | GNN* |
|---|---|---|---|---|---|---|---|---|---|---|---|
| **HCP-FI** | 0.284 | 0.288 | 0.283 | 0.287 | 0.281 | 6.548 | 0.278 | 0.290 | 0.282 | 0.283 | **0.264** |
| **HCP-WM** | 0.818 | 0.825 | 0.810 | 0.812 | 0.830 | 1.032 | 0.789 | 0.804 | 0.828 | 0.819 | **0.751** |

Table 6: Models' performance in terms of accuracy and MAE across five dynamic datasets

| Dataset | Accuracy ↑ | | | | | Dataset | MAE ↓ | | | | |
|---|---|---|---|---|---|---|---|---|---|---|---|
| | UniMP | $k-$GNN | GAT | SAGE | General | | UniMP | $k-$GNN | GAT | SAGE | General |
| DynHCP-Task | 89.66 | 73.03 | 89.67 | **90.93** | 68.84 | DynHCP-FI | **3.839** | 3.841 | 3.861 | 3.842 | 3.862 |
| DynHCP-Gender | **72.3** | 68.45 | 67.13 | 66.20 | 62.04 | DynHCP-WM | 10.589 | 10.596 | 10.592 | 10.597 | **10.571** |
| DynHCP-Age | **44.41** | 44.25 | 44.39 | 40.65 | 42.99 | | | | | | |

suited for graph machine learning techniques. To advance this vision, we present NeuroGraph, a comprehensive suite encompassing benchmark datasets and computational tools.

In our comprehensive exploratory study encompassing 35 datasets, we conducted a thorough analysis by running multiple machine learning models. Our key observations are as follows: (1) increasing the number of ROIs or employing large-scale brain graphs leads to improved performance compared to datasets with fewer ROIs, (2) employing a sparser graph setting enhances model performance and (3) while not intuitive, utilizing correlation as node features has significant potential to enhance model performance. Through a range of experiments across various learning objectives, we further highlight that GNNs exhibit superior performance compared to traditional NNs and 2D CNNs. These findings underscore the significant potential of GNNs in achieving improved performance across diverse tasks and underscore their suitability for graph-based Neuroimaging data analysis. Based on these insightful observations, we have developed NeuroGraph, a comprehensive benchmarks specifically designed for graph-based neuroimaging. Additionally, we provide computational tools to explore the design space of graph representation coming from Neuroimaging data, to facilitate the transformation of fMRI data into graph representations and showcase the potential of GNNs in this context. NeuroGraph serves as a valuable resource, offering a road map for researchers interested in leveraging graph-based approaches for fMRI analysis and demonstrating the effective utilization of GNNs in this domain.

## Acknowledgments and Disclosure of Funding

This research incorporates contributions from the "Modeling and Model Integration" project of the Vanderbilt Institute for Software Integrated Systems, and by the National Science Foundation under Grant No. 2325417, 2321684, 2239881 and NIH Grant No. RF1MH125931. Work on "Modeling and Model Integration" is supported by Wellcome Leap as part of the Multi-Channel Psych Program. Moreover, the data were provided [in part] by the Human Connectome Project, WU-Minn Consortium (Principal Investigators: David Van Essen and Kamil Ugurbil; 1U54MH091657) funded by the 16 NIH Institutes and Centers that support the NIH Blueprint for Neuroscience Research; and by the McDonnell Center for Systems Neuroscience at Washington University.

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

# A Benchmarks Availability and Licensing

The fMRI data utilized in this research was sourced from the Human Connectome Project [1]. The proposed graph-based benchmark datasets can be accessed for download at `https://anwar-said.github.io/anwarsaid/neurograph.html`. These datasets are provided in PyG[5] format, optimized for use with Graph Neural Networks (GNNs). However, they can also be conveniently incorporated into other platforms. Additionally, the associated code for downloading, preprocessing, and benchmarking is open to the public at `https://github.com/Anwar-Said/NeuroGraph`, complete with comprehensive documentation.

# B NeuroGraph and Neuroimaging Data

Neuroimaging, a powerful field of study, enables researchers to delve into the complexities of the human brain by capturing detailed images and measurements. Recent advancements in technology have resulted in an abundance of neuroimaging data, particularly functional magnetic resonance imaging (fMRI), which offers invaluable insights into brain activity. However, understanding and analyzing fMRI data pose several challenges. Firstly, the high dimensionality of fMRI data presents a significant hurdle. Additionally, inherent noise and variability in fMRI signals can obscure underlying neural activity. Complex spatial and temporal dependencies further complicate fMRI data analysis, demanding advanced modeling techniques. Furthermore, the interpretation and analysis of fMRI data can be time-consuming and subjective. The graphical representation of fMRI data offers a plethora of opportunities to tackle these challenges. For instance, network science and graph theoretical approaches provide a diverse range of tools to explore brain regions and their connectivity patterns [2]. Furthermore, the application of graph machine learning techniques, such as GNNs are particularly well-suited for analyzing neuroimaging data and have the potential to provide valuable insights. The provision of graph-based neuroimaging benchmarks and computational tools play a crucial role to enhance the field, which is the main focus of this study.

## B.1 fMRI Data Sources

Several initiatives have been undertaken in the past decade to assemble comprehensive fMRI datasets. One notable source is the Human Connectome Project (HCP) dataset [1]. The HCP dataset offers an extensive collection of multimodal neuroimaging data, including resting-state fMRI, task-based fMRI, and structural MRI scans, from a large cohort of healthy individuals. In addition to large neuroimaging datasets curated by institutions or projects, some notable resources are OpenNeuro, OpenfMRI and fcon_1000[6] platforms, which host a diverse range of publicly available fMRI datasets contributed by researchers worldwide [3, 4]. These datasets cover various experimental paradigms (see Table 7), clinical populations, and research domains, providing researchers with a wealth of data for analysis and investigation.

We have chosen to utilize the HCP S1200 dataset from the Brain Connectome as a primary resource for our graph-based benchmarking [1]. This dataset is well-suited for graph-based benchmarking due to its extensive coverage of brain regions and their interconnections. Additionally, the HCP S1200 dataset provides valuable demographic and behavioral information, enabling comprehensive analyses that consider various factors influencing brain connectivity. Its wide availability and standardized processing pipelines further contribute to its suitability for graph-based benchmarking, ensuring consistency and comparability across studies. Thus, the HCP S1200 dataset from the Brain Connectome represents a robust choice for conducting graph-based benchmarking studies in the field of neuroimaging.

## B.2 Reading HCP Dataset

Storing and reading fMRI datasets presents a formidable challenge due to their substantial storage requirements, necessitating significant disk space allocation, e.g., each subject of HCP S1200 requires 1.1 GB of space on disk. Moreover, the preprocessing of fMRI data calls for tools that are not only user-friendly but also highly efficient. Fortunately, the Human Connectome Database (HCP) offers

---

[5]`https://pyg.org/`
[6]`http://fcon_1000.projects.nitrc.org/`

Table 7: Description of fMRI paradigms in HCP Young Adult dataset.

| Dataset | Volumes per run (TR) | Run duration (min) | Duration of task blocks (sec) | Description |
|---|---|---|---|---|
| Rest | 1200 | 14 : 24 | - | no stimuli |
| Working Memory | 405 | 5 : 01 | 25 | 0-back, 2-back |
| Gambling | 253 | 3 : 12 | 28 | win, loss |
| Motor | 284 | 3 : 34 | 12 | various body parts |
| Language | 316 | 3 : 57 | approx. 30 | story, math |
| Relational Processing | 232 | 2 : 56 | 16 | relational, control |
| Social Cognition | 274 | 3 : 27 | 23 | interaction, no interaction |
| Emotion Processing | 176 | 2 : 16 | 18 | face, shape |

Table 8: fMRI scans required disk storage. The storage information is obtained from Human Connectome Project website.

| Task | Storage (GB) |
|---|---|
| Rest | 1260.95 |
| Working Memory | 527.70 |
| Gambling | 387.38 |
| Motor | 415.81 |
| Language | 426.72 |
| Relational | 343.40 |
| Social | 386.76 |
| Emotion | 295.91 |

an AWS instance (s3 bucket) that allows for seamless data crawling. NeuroGraph, with its implementation utilizing the boto3 Python package, provides an efficient solution for crawling the dataset. Boto3, a widely used Python package, enables seamless interaction with AWS services, facilitating efficient data retrieval and preprocessing in the NeuroGraph framework. Our implementation offers users the flexibility to either store the datasets or preprocess them on the fly if storage space is limited (see Table 8 for disk storage). To access the HCP data, users are required to obtain credentials from HCP[7] and provide them to NeuroGraph. Moreover, NeuroGraph also provides a Python class for preprocessing data from the local storage.

## B.3 Data Preprocessing

In close collaboration with domain experts from both the neuroimaging and graph machine learning fields, NeuroGraph's preprocessing pipeline is divided into five stages. These stages ensure the quality and reliability of the fMRI data. Initially, we utilize data that has already been processed using the HCP minimal processing pipeline [5].

- **Step 1 - Brain Parcellation:** The first phase of our pipeline involves brain parcellation, a process that divides the brain into smaller regions or parcels. This step allows for the analysis of functional connectivity within and between these parcels. In our study, we employ the Schaefer atlases [6], widely used brain parcellation schemes that define neurobiologically meaningful features of brain organization. These atlases provide a parcellation of the cerebral cortex into hierarchically organized regions at multiple resolutions. Using the population level atlases, we extract the mean fMRI timeseries for each region of interest (ROI). This provides a representative measure of the average neural activity within each specific brain region, enabling subsequent connectivity analyses.

- **Step 2 - Remove Scanner Drifts and Motion Artifacts:** Next, we remove linear and quadratic trends along with six rigid-body head motion parameters and their derivatives, from the fMRI data. Removal of the trends aims to remove the scanner drifts in the fMRI signals that arise from instrumental factors. Removal of the motion parameters, that capture the movement and rotation of the subject's head during the scanning session, ensures that any potential confounding effects are minimized. By eliminating these artifacts, we enhance the signal-to-noise ratio and increase the sensitivity to neural activity.

- **Step 3 - Subject-Level Signal Normalization:** We perform subject-level normalization of the ROI timeseries signals. Specifically, we temporally normalize all subject signals to zero mean and unit variance. This step allows for fair comparisons and facilitates the identification of meaningful variations in the functional connectivity patterns across subjects.

---

[7]https://db.humanconnectome.org

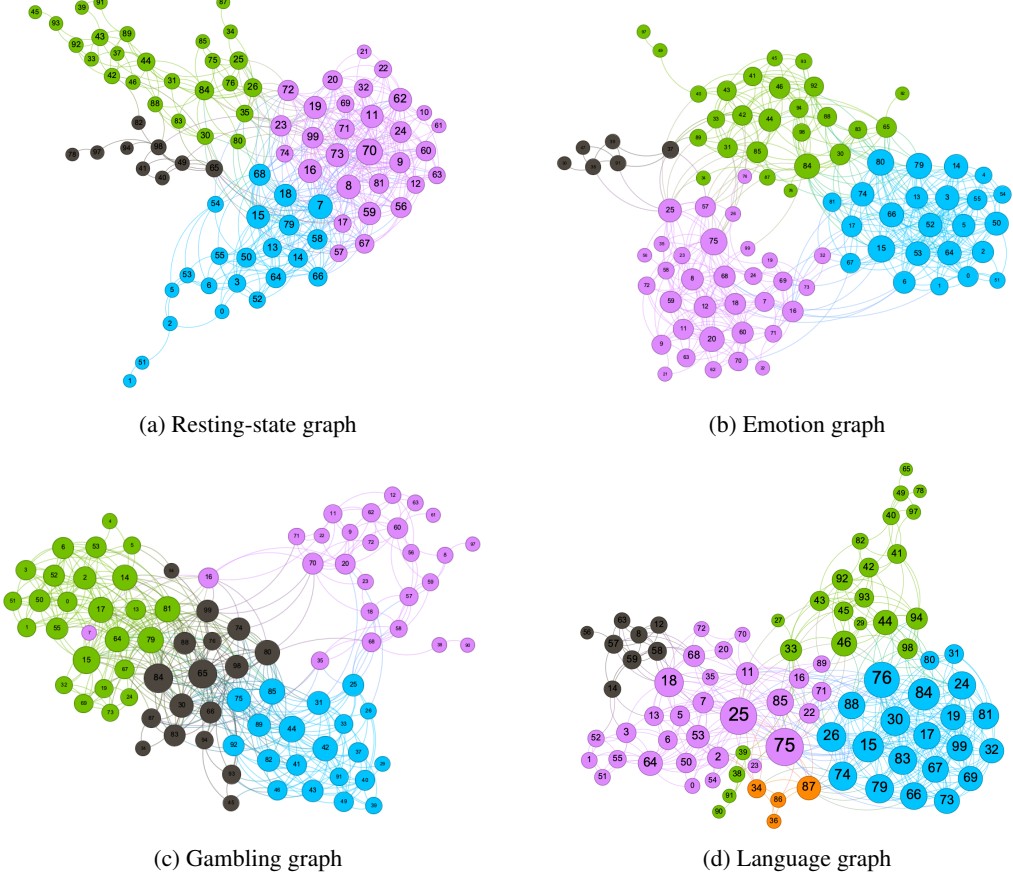

(a) Resting-state graph

(b) Emotion graph

(c) Gambling graph

(d) Language graph

Figure 3: Visualization of the corresponding simple undirected graphs with 100 ROIs for a single subject during both the rest condition and while performing certain tasks. Note that the coloring of the graphs has been applied based on the community structure, but solely for visualization purposes. Isolated nodes were removed.

- **Step 4 - Calculate Correlation Matrix:** We compute the correlation matrices from the ROI timeseries signals. Correlation matrices capture the strength of functional connectivity between different ROIs. By calculating pairwise correlations between the timeseries signals of each ROI, we obtain a matrix that represents the interregional functional connections within the brain. This step allows us to quantify and analyze the patterns of functional connectivity across the entire brain, and construct a graph. The correlation matrices serve as a valuable tool for investigating the network-level organization of the brain and identifying regions that exhibit synchronous activity [7]. These matrices provide a representation of the functional architecture and can be further utilized for graph-based analyses, such as network characterization and identification of key brain hubs [7, 8]. In Figure 3, we provide the visualizations of the graphs correspond to one subject in certain conditions.

- **Step 5 - Construct Static/Dynamic Attributed Graphs:** Finally, we compute two types of graph-based datasets from the functional connectivity matrix: static and dynamic graphs. As discussed in Section 3 of the paper, the static graph is defined as $G = (\mathcal{V}, \mathcal{E}, X)$. Here, the node set $\mathcal{V} = \{v_1, v_2, \ldots, v_n\}$ represents ROIs, while the edge set $\mathcal{E} \subseteq \mathcal{V} \times \mathcal{V}$ denotes positive correlations between pairs of ROIs, as determined by a predefined threshold. The feature matrix is represented by $X^{n \times d}$, where $n$ symbolizes the total number of ROIs, and $d$ corresponds to the dimension of the feature vector. We explore the dataset generation search space by considering different numbers of ROIs, different thresholds, and node features to identify optimal parameters. The next section provides a comprehensive overview of the dataset construction search space.

Regarding the parameter setup for constructing our benchmark datasets, we opt for a sparse setup (top 5%) with 1000 ROIs for the HCP-Gender, HCP-Age, HCP-WM, and HCP-FI datasets. However, for the HCP-Task dataset, we reduce the number of ROIs to 400 in order to manage memory overhead. In the dynamic setting, we employ a sliding window approach with a fixed window length ($\Gamma$) set to 50 and a stride of 3. Considering memory constraints and computational overhead, we fix the dynamic length ($l$) to 150 and slide over the preprocessed timeseries matrix to construct dynamic graphs. For all dynamic graphs, we consider 100 ROIs and medium sparsity (top 10%). With this setting, the total number of dynamic graphs we obtain for each subject is $((l - \Gamma)/stride) + 1$.

## B.4 The Design Space is Vast

The design space for constructing graphs from correlation matrices is substantial, given the multitude of available methods. We can construct diverse graph types employing various strategies. For instance, some of the potential graph types to consider include simple undirected graphs as demonstrated in [9], weighted graphs[10], attributed graphs [11], and minimum spanning trees [12, 13], among others. Similarly, a range of parameters comes into play during this process, further expanding the design space for these constructions. These parameters include the number of ROIs, edge weights, density thresholding for edge selection, and node features, to name a few.

GNNs have shown considerable promise in handling attributed graphs, demonstrating their effectiveness in various domains [10, 14]. Attributed graphs, which include not only the graph topology but also node-level features, represent complex systems more accurately than simple graph. GNNs leverage these attributes to capture both local and global structural information, allowing for the development of more comprehensive graph representations. Considering the importance of attributed graphs, we opted to construct rich, brain attributed graphs.

**Node Features:** Traditional methods for representing node features in graphs include using coordinates [15], one-hot encoding [8], and mean activation [16, 15]. Coordinates serve to provide spatial information about the nodes, while one-hot encoding are used for categorical features, effectively distinguishing different node types. Mean activation, on the other hand, can give insights about the average level of a node's activity or influence. While these methods provide a base level of information, they may not fully capture the rich complexity inherent in many data structures, such as brain graphs. To address this, we explore more powerful ways of representing node features, including using correlation vectors, BOLD signals and the combination of both. Correlation vectors can encapsulate the relationship between different nodes, providing insight into the connectivity and interaction within the graph. BOLD signals, give information about changes in blood flow in the brain, which can be an indicator of neural activity. By combining both of them, we may enrich models with a wealth of information, thereby capturing the intricate details and relationships present in brain graphs.

**Number of ROIs:** ROIs in brain graph construction may significantly impacts the granularity and overall scope of the resulting graph. Using a smaller number of ROIs, such as 100, can lead to a more generalized and coarser view of brain connectivity. This simplified perspective can be useful for broad overviews and initial exploration but might overlook intricate local interactions or specific clusters of activity. Conversely, using a larger number of ROIs, such as 400 or 1000, allows for a more detailed and finer representation of the brain's connectivity. With more ROIs, the graph can capture more specific interconnections, potentially revealing sub-networks or localized activity patterns that a coarser graph might miss. However, larger graphs also present a challenge in terms of computational load and complexity, also prone to noise. Interestingly, different methods in the literature have adopted different numbers of ROIs for their analysis [9, 10, 11]. These varying approaches underscore the fact that the choice of ROIs number is not merely a matter of computational convenience, but can significantly influence the outcomes of the study.

In light of this, our research aims to explore these three ROIs sizes: 100, 400, and 1000. Our goal is to understand the impact of different graph granularity levels on the performance of GNNs. By doing so, we hope to provide deeper insights into how different levels of detail in the graph structure affect the GNN's ability to capture and model brain connectivity. This investigation could potentially guide the selection of an optimal ROI size in future brain graph studies, striking a balance between capturing sufficient detail and maintaining computational feasibility.

**Density Thresholding:** Graph density is a fundamental property that may impacts the performance of GNNs. Graph density refers to the proportion of the possible connections in a graph that are actual connections. It influences how information is propagated through the network, may potentially affect the accuracy and efficiency of the GNN. A sparse graph (low-density) might lead to information underflow, with some nodes being poorly connected, which might cause inadequate learning of node representations. On the other hand, a dense graph (high-density) could lead to an information overflow, with a significant amount of information being propagated, possibly causing noise and overfitting [17].

Thresholding, on the other hand, is a crucial step in the construction of brain graphs. It's used to determine which correlations are strong enough to be included as edges in the graph. There are several approaches to thresholding. One is absolute thresholding, where a fixed threshold value is selected, and all correlations in the matrix above this threshold are included as edges in the graph. However, the choice of an absolute threshold can be somewhat arbitrary, and may result in graphs of varying sizes and densities. This variability can complicate comparisons between graphs [18]. Proportional thresholding is another method, in which the strongest $x\%$ of correlations are included as edges in the graph. This method ensures that all resultant graphs have the same density of edges, which facilitates comparisons between them. However, it can also result in the inclusion of weak, potentially non-significant correlations in the graph. To avoid this issue, some studies consider only positive correlations, which allows the construction of graphs with various densities and avoids the complications of negative thresholding [19].

Indeed, there are numerous ways to conduct thresholding in brain graph construction, with several options available within each thresholding approach. Each method and option presents its unique set of advantages and potential limitations. In this context, we focus on proportional thresholding with positive correlations, an approach that has shown encouraging results in previous research [10, 9]. Specifically, we explore three levels of density: those defined by the top 5%, 10%, and 20% percentile values from the correlation matrices. These densities represent different levels of graph sparsity, offering a broad perspective on how the choice of threshold can impact the topology and interpretability of the resulting brain networks. We note that the terms "sparse" (5%) and "dense" (20%) are relative and dependent on the context of feasible ranges. Despite their different percentages of edges, both sparse and dense graphs exhibit a complexity of $O(n^2)$ edges. We observed that even in sparse datasets, the average degree is around 50 for 1000 ROIs, indicating a substantial level of connectivity.

## C  NeuroGraph Benchmark Datasets

We propose a collection of ten datasets tailored to five distinct tasks, encompassing both static and dynamic contexts. These tasks are identified as HCP-Task, DynHCP-Task, HCP-Gender, DynHCP-Gender, HCP-Age, DynHCP-Age, HCP-WM, DynHCP-WM, HCP-FI, and DynHCP-FI. These datasets are derived from the HCP S1200 dataset, following a sequence of preprocessing operations. For the creation of static datasets, we eliminated two subjects that contained fewer than 1200 scans and then applied the preprocessing as outlined in the previous sections. The resulting datasets are represented as sparse matrices with 1000 ROIs. However, we've tailored the Activity dataset to include only 400 ROIs owing to its larger size of over 7000 scans, as this adaptation was necessary to overcome memory constraints. As for the dynamic datasets, we've standardized the dynamic length to 150, with a window size of 50 and a stride of 3. Moreover, to alleviate the substantial memory demands, we've limited the dynamic datasets to encompass only 100 ROIs. The distribution of classes for each dataset, as well as the values for regression tasks, have been visualized and are presented in Figure 4.

### C.1  GNN* and Dynamic Graph Baselines

Our study also explores a variation of residual GNNs, we named GNN*, the model that leverages both residual connections and a feature concatenation approach, enhancing the utilization of the functional connectome in the training process. As delineated in Section 3.4 and visualized in Figure 2 of the main paper, GNN* employs a universal graph convolution layer, facilitating the use of any GNN convolution contingent on the project's requirements. Similarly, the dynamic graph baseline (depicted in Figure 2 of the main paper) also uses a general graph convolution, followed by a Transformer

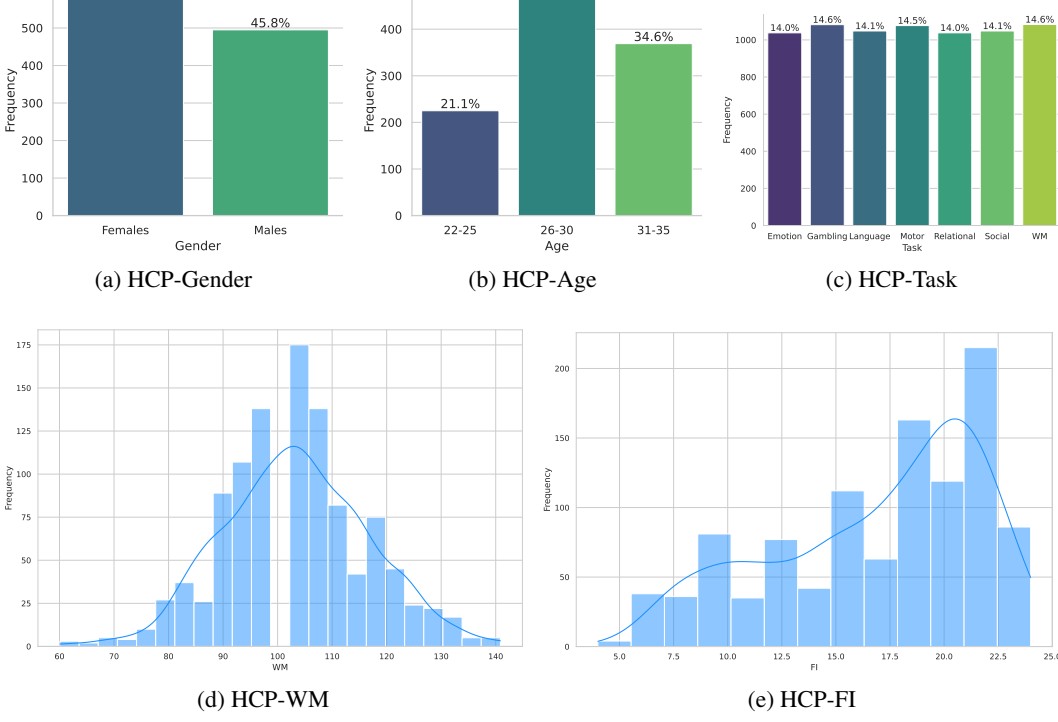

Figure 4: Illustration of class distribution for each dataset. For the regression task, histograms are presented to depict the frequency distributions of both Working Memory (WM) and Fluid Intelligence (FI) scores. In addition to these, Kernel Density Estimates are superimposed on the histograms, providing a smoother representation of the distributions.

module. Throughout our experimentation, we employed UniMP with GNN* and tested five models using the dynamic baseline, the results of which are tabulated in Table 6 of the main paper. All other parameters remain consistent with the detailed exposition in the experimental setup (Section 5.1) of the main paper.

# D  Memory and Running Time Analysis

Following a comprehensive and rigorous exploration of the search space, we have identified and established optimal datasets that strike a balance between minimizing memory requirements and maintaining an effective quantity of parameters. The trade-off achieved ensures that models are able to run smoothly on machines with reasonable computing power on our datasets, making them highly accessible to a wide range of users. This optimization also yields the additional benefit of reduced training times; our models are capable of training in mere minutes, significantly accelerating the model development cycle and promoting rapid iterative progress.

The specifics of this optimization are illustrated in the context of Unified Message Passing (UniMP) model [20], which we use to showcase the efficient resource usage of our datasets and approach. In Table 9, we offer detailed insights into the running times and memory requirements of UniMP model. We executed UniMP on each dataset for 100 epochs and recorded both GPU memory utilization and overall training time, which includes data loading. The number of hidden units for the GNN layer was 32 and 128 for the MLP layers. These data points provide a tangible representation of the efficiency gains achieved through our dataset size optimization process. Such optimizations are instrumental in ensuring datasets are not only computationally effective using any model but also highly accessible, enabling broader applicability for a variety of hardware configurations. All experiments were executed on a system equipped with an Intel(R) Xeon(R) Gold 6238R CPU operating at 2.20GHz with 112 cores, 512 GB of RAM, and an NVIDIA A40 GPU with 48GB of memory.

Table 9: Resource utilization analysis of UniMP model on all benchmark datasets

| Benchmark Dataset | Disk storage (GB) | #Parameters | Memory (MB) | Training time (sec) |
|---|---|---|---|---|
| HCP-Task | 4.0 | 265035 | 2463 | 854 |
| HCP-Gender | 3.7 | 648870 | 6437 | 362 |
| HCP-Age | 3.6 | 648903 | 4293 | 355 |
| HCP-WM | 3.7 | 803461 | 6551 | 696 |
| HCP-FI | 3.6 | 803461 | 6762 | 690 |
| DynHCP-Task | 7.3 | 309575 | 15881 | 11200 |
| DynHCP-Gender | 1.1 | 308930 | 4169 | 1700 |
| DynHCP-Age | 1.0 | 309059 | 4113 | 1709 |
| DynHCP-WM | 1.1 | 308801 | 4359 | 1704 |
| DynHCP-FI | 1.0 | 308801 | 4335 | 1712 |

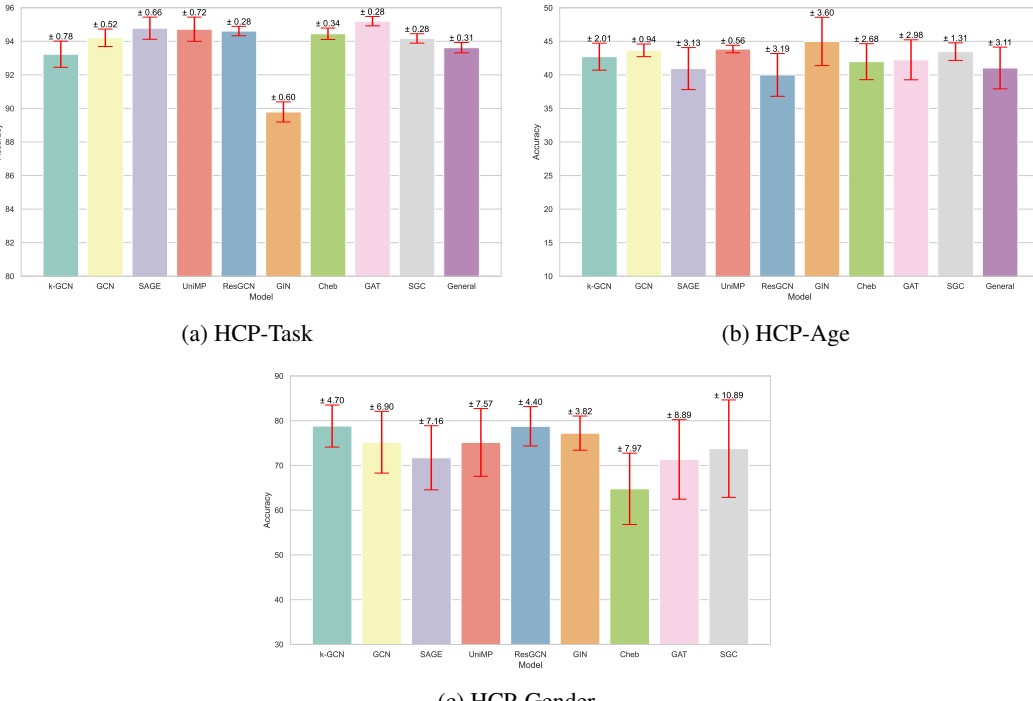

(a) HCP-Task

(b) HCP-Age

(c) HCP-Gender

Figure 5: Models' performance: Accuracy and standard deviation on 10 runs with different seeds on HCP-Task, HCP-Age and HCP-Gender datasets.

# E    Models Performance and Standard Error

We plot the accuracy along with the standard deviation of 10 runs, each with different seeds, for all the models on three distinct datasets: HCP-Task, HCP-Age and HCP-Gender in Figure 5. We observed that the results reported a higher level of stability on both HCP-Task and HCP-Age datasets. This indicates that the models performed consistently and yielded more reliable results, suggesting a greater degree of confidence in the accuracy measurements. On the HCP-Gender dataset, we observed slightly high standard errors across the models. Moreover, we provide the visualization of the hidden activations obtained from the last layer of $GNN^*$ for the test and validation sets trained on HCP-Task and HCP-Gender datasets in Figure 6. We used TSNE for these visualizations.

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
