# OpenReview forum: "NeuroGraph: Benchmarks for Graph Machine Learning in Brain Connectomics"
_NeurIPS.cc/2023/Track/Datasets_and_Benchmarks — NeurIPS 2023 Datasets and Benchmarks Poster_

### Official Review · Reviewer_4Rsw · 2023-07-08
**Lots of clarifications required from authors**

**Rating:** 6
**Confidence:** 3

**Strengths:**

1. Dealt with both static and dynamic graphs.
2. Clear conceptual explanation.
3. References are provided for all the technical explanations.


**Additional Feedback:**

English can be improved.

**Clarity:**

A correlation between paragraphs can be maintained. A few improvements that can be made are –
1.  In the introduction section, a paragraph mentioning different sections of the paper can be added.
2. There was no clear correlation between subsequent paragraphs.


**Correctness:**

The evaluation methods and experimental design employed in this study appear to be appropriate.

**Documentation:**

The given dataset is available publicly, and all details are given.

**Ethics:**

No issues

**Limitations:**

1.	1.In Table -1 nomenclature of all symbols is not mentioned.
2.	Figure-1 is not referred in the explanation.
3.	What represents edges in the generated graph.
4.	what are the challenges of the existing study?
5.	What are the six head motion parameter which are regressed out?
6.	They haven’t mentioned the name of which GNN model performed well for all types of variations.
7.	Correlation as a node features [a,b], and Sparser graph[c] are already used in these papers to get better  GNNs performance-
 a. Li, Xiaoxiao, et al. "Braingnn: Interpretable brain graph neural network for fmri analysis." Medical Image Analysis 74 (2021): 102233.
b. Ding, Kexin, et al. "Graph convolutional networks for multi-modality medical imaging: Methods, architectures, and clinical applications." arXiv preprint arXiv:2202.08916 (2022).
c. Pina, Oscar, et al. "Structural networks for brain age prediction." International Conference on Medical Imaging with Deep Learning. PMLR, 2022.


**Opportunities For Improvement:**

Larger dataset can be used for more accurate and comprehensive benchmarking of this study.

**Relation To Prior Work:**

The contribution of this study is similar to already existing studies/research papers

**Summary And Contributions:**

This study introduces NeuroGraph, a carefully selected and comprehensive benchmark dataset for graph-based neuroimaging. This study also offers computational tools that convert FMRI datasets into static and dynamic graph representations. The study presents a framework based on Graph Neural Networks (GNNs) for learning from static and dynamic graphs in various tasks such as age prediction, sex classification, and activity classification. The potential of GNNs is evaluated by comparing their performance with other baseline models.
The study reveals that a combination of correlation-based node features, a large number of Regions of Interest (ROIs), and a sparser graph leads to superior results.

---

> ### Author Response · Authors · 2023-08-22
> **Rebuttal by Authors**
>
> We appreciate the reviewer for carefully reviewing our paper and appreciating our idea. Below, we present our responses to each of the reviewer's comment.
>
> **Question 1:** In Table -1 nomenclature of all symbols is not mentioned.
> Figure-1 is not referred in the explanation.
>
> **Response:** We thank the reviewer bringing this to our attention. We have now revised the paper and provided detail for each notation accordingly. We have also referred Figure 1 in the text.
>
> **Question 2:** What represents edges in the generated graph.
>
> **Response:** Following to the widely recognized threshold-based graph generation methods, we add edge between two ROIs, each depicted as nodes within the corresponding graph, when their correlation exceeds a specific threshold. Our investigation encompasses three distinctive thresholding scenarios: the top 5\%, which we denote as sparse graphs; the top 10\%, referred to as medium; and the top 20\% positive percentile values, classified as dense graphs. A comprehensive examination of the effectiveness of each of these thresholding approaches is provided in section 5.2 and Table 3 of the main paper.
>
> **Question 3:** what are the challenges of the existing study?
>
> **Response:** This is an intriguing question. As detailed in Supplementary Material Section A.4, the design space for generating graph-based brain datasets is vast. It encompasses numerous hyperparameters, encompassing diverse preprocessing pipelines, a range of parcellation methods, and the potential for varying graph data modalities and thresholding. While this necessitates detailed investigation, it's unfortunate that this domain remains relatively unexplored at present. Our current study delves into several pivotal aspects, yet we acknowledge that there exists considerable uncharted territory. Furthermore, the process of graph generation itself results in information loss as it ignores correlations between ROIs that fall below the defined threshold. Another significant limitation of this study is the limited exploration and evaluation of dynamic datasets, warranting further investigation in this domain. We are of the opinion that our tool, NeuroGraph, will facilitate the exploration of these directions and encourage further advancements.
>
> We mentioned some of the limitations in the supplementary material, however, we have now added a paragraph (posted above, **Limitations and Future Work** )to the main draft highlighting the limitations and potential avenue for future research.
>
> **Question 4:** What are the six head motion parameter which are regressed out?
>
> **Response:**  The first six variables are the motion parameters estimates from a rigid-body transformation to a reference image acquired at the start of each fMRI scan, which are translation x,y,z (in mm) and rotation x,y,z (in deg). Details can be found in *WU-Minn HCP 1200 Subjects Data Release Reference Manual.*
>
> **Question 5:** They haven’t mentioned the name of which GNN model performed well for all types of variations.
>
> **Response:** We thank the reviewer pointing this out. The experimental results demonstrate that the utilization of residual connections in GNNs (as shown in Figure 2 (b), Table 4 and 5) consistently leads to improved performance. We plan to include more explanation to the additional content page in the revised version.
>
> **Question 6:** Correlation as a node features [a,b], and Sparser graph[c] are already used in these papers to get better GNNs performance-
>
> **Response:** We agree with the reviewer's observation that correlations have been employed using both node and edge features in certain prior studies, as indicated in the reference provided. However, it's worth highlighting that an in-depth evaluation of different node features and an extensive exploration of the associated design space remain unexplored. To the best of our knowledge, our study represents the first effort in this direction. Furthermore, our contribution extends beyond the exploration itself. We present graph-based benchmarks, complete with user-friendly implementations tailored for graph machine learning applications. Moreover, our contribution encompasses a Python tool designed for fMRI preprocessing and the construction of graph-based datasets, effectively bridging the gap between these two distinct domains.
>
> **Question 8:** A correlation between paragraphs can be maintained. A few improvements that can be made are –
> * In the introduction section, a paragraph mentioning different sections of the paper can be added.
> * There was no clear correlation between subsequent paragraphs.
>
> **Response:** We thank the reviewer bringing this to our attention. We will made the suggested changes to the camera-ready version.

---

> > ### Comment · Reviewer_4Rsw · 2023-08-29
> > **Thanks for your replies**
> >
> > The replies seems reasonable however I would not change my rating..Please program chairs consider  all reviewer comments and decide about acceptance...Best wishes

---

> > > ### Author Response · Authors · 2023-08-29
> > > **Thank you**
> > >
> > > We appreciate the reviewer taking the time to review and confirm our response.

---

### Official Review · Reviewer_g2aw · 2023-07-21
**Review for "NeuroGraph: Benchmarks for Graph Machine Learning in Brain Connectomics"**

**Rating:** 6
**Confidence:** 3

**Strengths:**

From my point of view, the main strengths of the paper are:

-	Considering the standard size of MRI datasets, the datasets compiled in the current paper have a large sample size, which enables the usage of novel GML methods, such as the ones outlined throughout paper.

-	The benchmarking carried out by the authors is quite complete. They have tested many methods on different configurations of the data, in terms of feature and graph dimensionality.

-	The authors achieve good results in many tasks, showing the potential of the data and the methods tested.

-	Besides, the authors present a competitive GNN architecture that proves also quite performing in many of the tasks.

**Additional Feedback:**

I don't have any more feedback on the manuscript.

**Clarity:**

Yes, the paper is well written. It is easy to follow, and the structure is correct, making it easy to understand. Besides, the supplementary material presents some additional important information, such as analysis of memory and runtime, class distribution for some of the tasks, etc.

**Correctness:**

Yes, the authors explain all the assumptions for the construction of the datasets quite clearly, and the importance of each decision taken. Besides, the authors perform an interesting benchmarking on a broad set of methods.

**Documentation:**

Yes, the information is quite complete. The repository has really well-detailed documentation, with details on the preprocessing tools, the data, benchmarks, installation, etc. Therefore, I have no doubt it should be possible to easily install everything, load the data, and replicate the experiments, though I did not try. Besides, they specify a license and information on how to cite them.

**Ethics:**

No, they utilize data from HCP, which has gone already through all the required steps of anonymization, etc.

**Limitations:**

I don’t think this work poses any potential negative societal impact. Regarding the limitations listed before, I feel the authors could have obtained more complete datasets from the original HCP data, as discussed before.

**Opportunities For Improvement:**

Even though the authors present good and interesting results, I feel the datasets presented are quite limited, and if these end up being extensively used, they could limit the possibilities of the techniques benchmarked. The main reason for this statement is that I feel the processing done, on top of the minimal preprocessing outlined by the HCP recommendations, is quite limited, and the node features captured do not fully exploit the possibilities of the source fMRI data. More specifically, I have the following concerns:

-	It seems the most discriminative features are the correlation values. However, these should implicitly be captured in the graph edges, and I feel it is quite redundant to have them also as node features.

-	The raw BOLD signal I feel is quite complex and high dimensional to be harnessed efficiently by the methods utilized. However, no other features are extracted from it, which hinders understanding how this signal can contribute to better solving the confronted tasks.

-	In fMRI analysis, it is quite extended the usage of ICA to extract independent brain networks, i.e. subnetworks of specific ROIs that communicate in a certain manner to each other, to coordinate concrete tasks. Despite this would have been a great addition to the features extracted, and would have also allowed the generation of sparser sub-graphs, this technique is not mentioned, utilized, or extracted.

-	Finally, many more features, specifically related to each ROI, could have been extracted, such as volumetric characteristics, level of activation, etc. Even though they may have ended up being useless, it would have allowed exploring a more overcomplete representation of the original data.

For all these reasons, I do see the paper more as a thorough benchmark of different GML methods, rather than a complete dataset for testing new GML methods for the analysis of fMRI data.

Beyond, I have some further comments on the manuscript:

-	I believe the claim of the authors on Page 2, line 63, “transform neuroimaging data into a standard graph representation suitable for GML”, is a bit exaggerated. As commented before, I have the humble opinion that the source data could have been better encoded, and more features could have been extracted. Therefore, making these datasets a standard could hinder the development of more complex GML methods that rely on richer features.

-	The authors pose “research questions” in Section 4. To be honest, I don’t think these should be named as research questions, but rather the results of hyperparameters’ selection. And the reason for that is that we cannot expect a single answer per question, but rather different values for the parameters depending on the method chosen.

-	From Table 3 the authors conclude that using sparse edges is the best configuration for consequent experiments. But in this table we still find that medium or dense schemas are also quite competent when performing activity classification.

-	In Table 4, are the results provided the balanced accuracy? It is not clear and will change the interpretation of the results.

**Relation To Prior Work:**

I feel that the authors comment more on previous techniques for analyzing the functional connectome, rather than discussing previous datasets in the neuroimaging community, and how they are built. And since the paper presents a new dataset, I believe they should be reflecting more on previous cases, and how their proposal is differentiated from these. Besides, I believe it could have been possible to present other approaches for encoding fMRI data, such as through the extraction of independent brain networks, or previous work on rich node features that they could have leveraged to further enhance the constructed graphs.

**Summary And Contributions:**

The authors propose a collection of datasets leveraging the HCP data for benchmarking graph machine learning approaches on functional MRI data. These datasets cover a large set of problems, such as age or task prediction, and different data representations, more specifically static and dynamic networks. The authors benchmark different sets of node features, degree of brain parcellations, and edge sparsity, concluding some guidelines for researchers interested in harnessing graph machine learning approaches.

---

> ### Author Response · Authors · 2023-08-22
> **Rebuttal by Authors**
>
> We thank the reviewer for reviewing our manuscript and providing valuable feedback. In the following, we respond to their comments.
>
> **Question 1:**  I feel it is quite redundant to have the correlation features also as node features.
>
> **Response:** We thank the reviewer for highlighting this point. We would like to note that the incorporation of correlations as edge features, which would typically involve a single correlation value, primarily captures the strength of correlation between two highly correlated ROIs.  This may offer a modest contribution to the learning. However, utilizing correlations as node features empowers the message passing neural network (MPNN) to extract insights from correlations among all ROIs, leading to a substantial performance enhancement [1]. In this context, we can envision the MPNN as an additional function to the MLP, facilitating the exchange of messages between highly correlated ROIs while leveraging the entirety of the connectome information. Moreover, with additional MLP and residual connections, we observed an overall improvements on all tasks, especially up to 6\% on  HCP-Age dataset. We'd also like to kindly point out that our eventual constructed dataset has only node features, and not weighted edges. Thus, in the graph representation, the information is not actually exactly redundant. However, we also acknowledge that considering edge features and weighted graph representations are potential avenues for future research. We hope that NeuroGraph will stimulate and support further advancements in this direction.
>
> **Question 2:** No other features are extracted from BOLD, ROIs and the usage of ICA would have been a great addition to the features.
>
> **Response:** We thank the reviewer for their extensive comments. As we detailed in our supplementary material (Section A.4), the landscape of constructing brain graphs encompasses a variety of approaches. Previous research in this domain has explored various methodologies for constructing brain graphs and subsequent downstream tasks. For instance, in [1,2,3,4], distinct measures such as mean activation, node index as coordinates, one-hot encoding, spatial one-hot encoding, and correlations have been employed as node features with brain graphs. Additionally, a diverse range of measures including partial correlation, Pearson correlation, and geometric distances, among others, have found wide application in defining edges within brain graphs. Moreover, the fMRI preprocessing pipelines before graph construction used in prior studies have high variability, and there is no consensus on the optimal combination of parameters that yield expressive graph datasets capable of enhancing the overall predictive capacity of graph machine learning approaches. Unfortunately, this design space is currently underexplored, with very few efforts made in this direction so far. We believe that the primary reason for this gap is the lack of tools and benchmarks that facilitate easy exploration of this design space. We also acknowledge that designing such tools presents a formidable challenge, as it necessitates expertise in both neuroscience and graph machine learning domains. Mutual collaboration between these domains is essential to address this limitation effectively.
>
> As the reviewer highlighted, there are numerous ways to construct distinct features from the functional connectome and BOLD signals. Notable examples encompass fALLF, ICA, DifuMo, among others. However, delving into the expansive design space and exploring all possible combinations of parameters is a challenging task. To navigate these challenges, this study serves as a foundation by introducing a Python tool, NeuroGraph for preprocessing fMRI data and constructing brain graphs. It's worth acknowledging that while we address a pivotal subset of hyperparameters, we recognize that the design space is far-reaching. Furthermore, it’s important to highlight that this study aimed not only to offer graph-based neuroimaging benchmarks, but also to introduce a Python tool featuring user-friendly interface for constructing brain datasets. As previously mentioned, the potential design variations are extensive, we continue to actively explore the search space and refine the tool. Furthermore, we hope that NeuroGraph and proposed benchmarks promote further investigations into these questions. We have added a paragraph (posted above in response to reviewer awZz) to the main draft that provides these points as limitations/potential future research directions. Moreover, we will also rephrase our claim on the standard representation within the main draft.

---

> > ### Author Response · Authors · 2023-08-22
> > **Rebuttal by Authors**
> >
> > **Questions 3:**  The authors pose “research questions” in Section 4. To be honest, I don’t think these should be named as research questions, but rather the results of hyperparameters’ selection. And the reason for that is that we cannot expect a single answer per question, but rather different values for the parameters depending on the method chosen.
> >
> > **Response:* We concur with the reviewer that these indeed pertain to hyperparameter questions. However, the answers to these hyper parameter questions also hold implications for neuroscience, making them also research questions. For instance, the determination of the number of regions pertains to the spatial resolution of these phenomena. The process of choosing the spatial granularity aligns with highly relevant questions in neuroscience.
> >
> >
> > **Question 4:**  From Table 3 the authors conclude that using sparse edges is the best configuration for consequent experiments. But in this table we still find that medium or dense schemas are also quite competent when performing activity classification.
> >
> > **Response:** We acknowledge the reviewer's observation regarding the varied results presented in Table 3. We would like to note that sparse graphs yield superior results across the majority of cases. For instance, based on the results from Table 3, the ratio of sparse:medium:dense on the gender classification dataset is 8:0:2, while on the activity dataset, it stands at 4:2:4. Furthermore, the differences in results, especially in cases where sparse graphs exhibit lower performance, are generally small. Recognizing the increased complexities stemming from memory usage, training demands, and the possibility of oversmoothing, we have chosen sparse graphs.
> >
> > We thank the reviewer highlighting this point. We will revise the paper to elucidate this rationale.
> >
> > **Question 5:** In Table 4, are the results provided the balanced accuracy? It is not clear and will change the interpretation of the results.
> >
> > **Response:** We use simple accuracy throughout our experimental setup. In our supplementary material, specifically Figure 3, we show the class distribution for each dataset. Notably, the HCP-Gender and HCP-Activity datasets exhibit a relatively balanced distribution, while some degree of imbalance is evident in the HCP-Age dataset, although not to a considerable extent. Our confidence in the appropriateness of accuracy as an evaluation measure remains consistent.
> >
> > **Question 6:** I feel that the authors comment more on previous techniques for analyzing the functional connectome, rather than discussing previous datasets in the neuroimaging community, and how they are built.
> >
> > **Response:** We thank the reviewer bringing this to our attention. We will expand section 2 of the main paper on the additional content page and discuss previous neuroimaging datasets and their preprocessing pipelines.
> >
> > [1]. Kim, Byung-Hoon, Jong Chul Ye, and Jae-Jin Kim. "Learning dynamic graph representation of brain connectome with spatio-temporal attention." Advances in Neural Information Processing Systems 34 (2021): 4314-4327.
> >
> > [2]. Li, Xiaoxiao, et al. "BrainGNN: Interpretable brain graph neural network for fmri analysis." Medical Image Analysis 74 (2021): 102233.
> >
> > [3]. Li, Xiaoxiao, et al. "Graph neural network for interpreting task-fmri biomarkers." Medical Image Computing and Computer Assisted Intervention–MICCAI 2019: 22nd International Conference, Shenzhen, China, October 13–17, 2019, Proceedings, Part V 22. Springer International Publishing, 2019.
> >
> > [4]. Kim, Byung-Hoon, and Jong Chul Ye. "Understanding graph isomorphism network for rs-fMRI functional connectivity analysis." Frontiers in neuroscience 14 (2020): 630.

---

> > > ### Comment · Reviewer_g2aw · 2023-08-28
> > > **Answers to question 3 to 6, and review of score**
> > >
> > > After my previous response, I would like now to answer the remaining questions. As stated before, I am also satisfied with your answers, and the modifications you are going to carry out in the paper. Still, I am not fully convinced by Q3, but in any case, this is really a minor thing. I am similarly not fully convinced by the answer to Q4, but in the light of the results presented on Table 3, I completely understand how proceeding with sparse graphs is the way to go, the most sensible one. Finally, I appreciate the answers to Q5 and Q6, as they fully clarify my concerns.
> > >
> > > For all these reasons, after having cleared up some of the concerns I had with the paper, and really understood the possibilities of the Python tool presented, NeuroGraph, I have decided to revise my score from 4 to 6. I am sorry for not considering a higher score, but I do still feel the authors could have been more exhaustive in some of the discussed points.
> > >
> > > Thank you so much for the dedicated time.

---

> > > > ### Author Response · Authors · 2023-08-28
> > > > **Thank you for acknowledging our response!**
> > > >
> > > > We greatly appreciate the reviewer taking the time to provide us with insightful feedback on the paper and the rebuttal. We are pleased that the reviewer has agreed with our response and understands the challenge of exploring the overall design search space. We also thank the reviewer for increasing their score.
> > > >
> > > > In response to Q3, as the reviewer hasn't been fully convinced by the use of the "research questions" terminology, and considering it's a minor concern as highlighted by the reviewer, we will revise our terminology and change it to "hyperparameter probe/exploration." We also appreciate the reviewer's understanding of our summarized observation in Q4 concerning sparsity.

---

> > ### Comment · Reviewer_g2aw · 2023-08-28
> > **Thanks for the responses**
> >
> > Dear authors,
> >
> > Thank you so much for the extensive responses. I am quite satisfied with your answer. First, I agree and understand the great benefit of adding the correlation as node features, as you clearly show in all your benchmarks. Regarding the second question, I completely understand the impossibility of performing an exhaustive feature exploration. For that reason, I appreciate you have clarified how the Python tool you are providing could actually help other researchers extending the study, and using other features. Therefore, I feel you have clearly answered these concerns. Thank you so much for the time taken.

---

### Official Review · Reviewer_awZz · 2023-07-24
**A useful resource for graph machine learning researchers**

**Rating:** 7
**Confidence:** 3

**Strengths:**

The authors provide a path for machine learning researchers to offer new tools to the neuroimaging community without delving into the detail of extracting data from fMRI scans. This work takes some of the guess work out of signal extraction, processing, and building the graphs that researchers will ultimately need. Running existing GNNs against this data further provides researchers interested in this field (or graph learning) in general a great starting point. The supplemental material does an excellent job introducing the underlying data.

**Additional Feedback:**

GNNs perform better on sparser graphs because most GNN's suffer from heterophily and over-smoothing. Newer topological approaches seek to overcome these limitations and this dataset could be an intriguing one for that line of research.

**Clarity:**

The paper is difficult to follow. There is alot to cover here, but I think it would it be helpful to blend some of the material from the supplemental into the main body of the text and better setup the design decisions for constructing the graphs.

**Correctness:**

While I have concerns over the design space being so large (as the authors point out in the supplemental), I don't see anything glaringly wrong in the way the dataset is constructed. I'm a little concerned about the OOM errors in Table 3 and wonder what the limiting factor is there.

I would also ask the authors to take a look at the results in Table 3 for the Chebyshev Convolution model on activity classification, 400 ROIs, dense. It seems oddly low compared to the rest of the values in that column, row, and the table as a whole.

**Documentation:**

Yes, the documentation is appropriate and available.

**Limitations:**

I do believe the authors could do a better job of explicitly calling out the limitations of their work. I am particularly concerned as to whether there are issues with encoding rich neuroimaging data in a graph (as expressive as they may be). Because the graph defines the structure, there is a real risk of information loss and introduction of bias that I do not see discussed in the current version of the manuscript. Some of this is covered in the supplemental but it would be good to have this in the main body of the text.

**Opportunities For Improvement:**

While the split between "dynamic" and "static" makes some intuitive sense, the authors could do a better job handling that explanation.

**Relation To Prior Work:**

The authors should address BrainGB and do a better job citing literature for existing GML and neuroimaging. It has been a busy 2022/2023 in that field.

**Summary And Contributions:**

The authors hope to strengthen the connection between graph machine learning and neuroimaging by creating a benchmark for comparison. They take existing publicly available data from the Human Connectome Project and use existing preprocessing pipelines before extracting fMRI time series and splitting the data into "dynamic" and "static" graphs. These graphs contain multiple levels of ROIs and sparsity. The authors then select 10 GNNs to perform 2 classification tasks on the dataset and measure the results against a 3-layer NN, a 2D CNN and Random Forest. This dataset will no doubt be useful for researchers interested in overcoming current limitations with GNNs and developing new methods.

---

> ### Author Response · Authors · 2023-08-22
> **Rebuttal by Authors**
>
> We extend our gratitude to the reviewer for acknowledging the merit of our paper and for offering us invaluable insights. Kindly find our detailed response addressing each of the points raised below.
>
> **Question 1:** While the split between "dynamic" and "static" makes some intuitive sense, the authors could do a better job handling that explanation.
>
> **Response:** We thank the reviewer bringing this to our attention. We will expand both Section A.3 in the supplementary materials and section 3.2 in the main paper and provide more comprehensive explanations. Specifically, we plan to extend the discussion on constructing dynamic graphs, including details about the sliding window technique, dynamic length, and highlighting their differences.
>
> **Question 2:** I do believe the authors could do a better job of explicitly calling out the limitations of their work. I am particularly concerned as to whether there are issues with encoding rich neuroimaging data in a graph (as expressive as they may be). Because the graph defines the structure, there is a real risk of information loss and introduction of bias that I do not see discussed in the current version of the manuscript. Some of this is covered in the supplemental but it would be good to have this in the main body of the text.
>
> **Response:** We thank the reviewer for also bringing this to our attention. We have now added a paragraph to the manuscript (posted below) discussing the limitations/future directions of this work. We also make stronger reference to section A.4 in the supplementary material where we address these details in depth.
>
> **Question 3:** While I have concerns over the design space being so large (as the authors point out in the supplemental), I don't see anything glaringly wrong in the way the dataset is constructed. I'm a little concerned about the OOM errors in Table 3 and wonder what the limiting factor is there.
>
> **Response:** We thank the reviewer for bringing this to our attention. Given the extensive preprocessing computations required for constructing these datasets, we distributed the experiments across two machines. One of them had a limited 12GB GPU memory, which resulted in OOM errors. The other machine is equipped with a GPU with 48GB of memory. While we initially believed that we had re-run all the experiments that failed due to OOM on the smaller GPU using our larger GPU, it seems that a few were overlooked. Therefore, we have now successfully completed these experiments and included the results in the table. Our new results align with the previous findings, for instance, ResGCN is still the best model on dense graphs (1000ROIs).
>
> **Question 4:**  I would also ask the authors to take a look at the results in Table 3 for the Chebyshev Convolution model on activity classification, 400 ROIs, dense. It seems oddly low compared to the rest of the values in that column, row, and the table as a whole.
>
> **Response:** We greatly appreciate the reviewer for pointing this out. It was an oversight on our part, and we have rectified the typo from 47.24 to 87.24 in our results.
>
> **Question 5:** The paper is difficult to follow. There is alot to cover here, but I think it would it be helpful to blend some of the material from the supplemental into the main body of the text and better setup the design decisions for constructing the graphs.
>
> **Response:** As per reviewer's suggestion, we will expand section 3.2 of the main paper in the camera-ready version on the additional content page.
>
> **Question 6:** The authors should address BrainGB and do a better job citing literature
> for existing GML and neuroimaging. It has been a busy 2022/2023 in that field.
>
> **Response:** While we cited BrainGB (Reference \#7 in the paper) and provided a brief overview of related work due to limited space, we will elaborate on the literature, specifically adding details about BrainGB [1], BrainGNN [2], [3], and [4] in the camera-ready version.
>
> [1]. Cui, Hejie, et al. "Braingb: A benchmark for brain network analysis with graph neural networks." IEEE transactions on medical imaging 42.2 (2022): 493-506.
>
> [2]. Li, Xiaoxiao, et al. "BrainGNN: Interpretable brain graph neural network for fmri analysis." Medical Image Analysis 74 (2021): 102233.
>
> [3]. Ding, Kexin, et al. "Graph convolutional networks for multi-modality medical imaging: Methods, architectures, and clinical applications." arXiv preprint arXiv:2202.08916 (2022).
>
> [4]. Pina, Oscar, et al. "Structural networks for brain age prediction." International Conference on Medical Imaging with Deep Learning. PMLR, 2022.

---

> > ### Author Response · Authors · 2023-08-22
> > **Revision to the paper**
> >
> > **Limitations and Future Work**
> >
> > The design space for graph construction from fMRI is huge, since a variety of techniques can be employed to generate various forms of graphs. For instance, a few potential representations include simple undirected graphs [24], weighted graphs [29], attributed graphs [9], and minimum spanning trees [4,46], among others. Similarly, a number of parameters are involved in the design space, thereby expanding the possibilities for brain graph construction. These parameters encompass considerations such as the choice of parcellation methods, where various brain atlases can be employed to segment the brain into regions of interest. Equally crucial is defining the number of ROIs, as it wields a significant influence on exploring brain functions. The selection of ROIs count permits the creation of brain graphs of varying sizes, allowing the opportunity to focus on both the global and granular levels of the brain. Additionally, the selection of a connectivity metric (such as correlation, coherence, or mutual information) can shape the interactions captured by the graph, thereby opening another avenue for investigation. Distinct metrics may accentuate different facets of brain connectivity, giving rise to diverse interpretations. Furthermore, diverse preprocessing steps for fMRI data, including motion correction and artifact removal, can impact the final connectivity estimates, creating another separate avenue for future research. An array of other parameters, such as varying edge attributes, edge weights, density thresholds, and assorted approaches to incorporating node features, remain as wide-open paths for future exploration. *This work addresses only a fraction of the potential hyperparameters within this expansive design space, while the remaining design space is still an open question.* Moreover, the graph representation approach also leads to information loss due to ignoring correlations between ROIs that fall below the considered threshold. Additionally, we believe that current evaluations of dynamic graphs are limited and more experiments with varying dynamic graph learning architectures are necessary. To promote further research and explore this vast design space, this work provides a Python tool and benchmark datasets. *Our aspiration is that this benchmark and tool will stimulate further investigations into these pivotal questions.*"

---

### Official Review · Reviewer_Zh62 · 2023-07-25
**A novel benchmark paper with brain connectomics**

**Rating:** 7
**Confidence:** 3
**Correctness:** It is constructed in a sound way.
**Clarity:** Yes

**Strengths:**

1. They provide a pipeline extracting from neuroimage data to sparse graph data rather than just simple datasets. It can provide more datasets with the development of neuroimage datasets.

2. The provided datasets collect both static and dynamic graphs, which enables novel tasks such as gender/age classification and mental state decoding.

3. The authors provide an extensive exploratory study on the pipeline and provide insightful experimental analysis.


**Additional Feedback:**

N/A

**Documentation:**

There is sufficient detail

**Limitations:**

See *Opportunities For Improvement


**Opportunities For Improvement:**

1. Regions of Interest (ROI) lie in the core foundation of the dataset as the building nodes. Authors directly use the Schaefer atlases to find the ROIs. Are there other parcellation methods? I would like to see an explanation of why using this method.

2. In section 3.4 when benchmarking on the dynamic graphs, authors provide a transformer-based baseline to take the place of previously existing ones. Though it is a small part of the whole paper, I do not think it is a proper way to ignore previous methods. I think at least providing some runnable results would be better.


**Relation To Prior Work:**

Yes

**Summary And Contributions:**

In this paper, the authors provide a bunch of graphs extracted from the Human Connectome Project. It greatly enriches the graph types from brain connectomics and the following new tasks from the novel datasets such as gender classification and age classification. Authors treat regions of interest (ROIs) as nodes and build edges based on the highest connectivity between ROIs. Both static and dynamic graphs are provided, and the pip tools for utilizing the graph are also provided. In summary, it is a novel graph dataset in current literature, and I suggest an acceptance.

---

> ### Author Response · Authors · 2023-08-22
> **Rebuttal by Authors**
>
> We thank the reviewer for reviewing our manuscript and providing us valuable feedback. We are pleased to learn that the reviewer regards our paper as a novel contribution with potential significance to the community. Below, we provide responses to their comments.
>
> **Question 1.** Regions of Interest (ROI) lie in the core foundation of the dataset as the building nodes. Authors directly use the Schaefer atlases to find the ROIs. Are there other parcellation methods? I would like to see an explanation of why using this method.
>
> **Response:**  We thank the reviewer for highlighting this point. Yes, numerous brain atlases exist in neuroscience literature. Among these, the Schaefer atlas stands out for its robustness, functional connectivity basis and support of multiple resolutions, making it a popular choice for brain network analysis [1,2,3]. A recent study [2] even delves into the comparative performance of various atlases, including Schaefer, when used with diverse conventional machine learning methods. Although the efficacy of each atlas may vary across different datasets, the Schaefer atlas consistently delivers competitive results. Taking this into consideration, we have opted to employ the Schaefer atlas within our experimental setup. Furthermore, our pipeline can be readily extended to incorporate other atlases, or at least to replicate the robustness of the Schaefer atlas.
>
> **Questions 2:**  In section 3.4 when benchmarking on the dynamic graphs, authors provide a transformer-based baseline to take the place of previously existing ones. Though it is a small part of the whole paper, I do not think it is a proper way to ignore previous methods. I think at least providing some runnable results would be better.
>
> **Response:** We agree with the reviewer's point. However, the codebases available for dynamic GNNs in the graph classification and regression settings are not as advanced as other codebases. Reimplementing these codes require significant amount of time. For instance,  Kim et al., [3] introduce a framework for learning dynamic brain graph models. Yet, its implementation employs custom torch functions that require substantial effort for adaptation into the PyTorch Geometric (PyG) framework. Additionally, while BrainGB [4] offers a PyG-based platform, it is tailored for static brain graphs and cannot be directly applied to dynamic graphs like ours.
> Given these constraints, we have taken the initiative to create a generalized dynamic GNN architecture to enable the evaluation of different GNN convolutions. We plan to incorporate the implementation of our proposed generalized dynamic architecture, along with a few static GNNs, into the NeuroGraph package to allow address these challenges.
>
> *While we concur with the reviewer's perspective, we intend to acknowledge this as a limitation in our paper.*
>
>
> [1]. Xu, J., Bian, Q., Li, X., Zhang, A., Ke, Y., Qiao, M., ... \& Gulyás, B. (2023). Contrastive Graph Pooling for Explainable Classification of Brain Networks. arXiv preprint arXiv:2307.11133.
>
> [2]. Huang, David Tse Jung, et al. "Data-Driven Network Neuroscience: On Data Collection and Benchmark." arXiv preprint arXiv:2211.12421 (2022).
>
> [3]. Kim, Byung-Hoon, Jong Chul Ye, and Jae-Jin Kim. "Learning dynamic graph representation of brain connectome with spatio-temporal attention." Advances in Neural Information Processing Systems 34 (2021): 4314-4327.
>
> [4]. Cui, Hejie, et al. "Braingb: A benchmark for brain network analysis with graph neural networks." IEEE transactions on medical imaging 42.2 (2022): 493-506.

---

> > ### Comment · Reviewer_Zh62 · 2023-08-22
> > **Keep score unchanged**
> >
> > I am confirming the author's response. A score of 7 fits the paper

---

> > > ### Author Response · Authors · 2023-08-22
> > > **Thanks**
> > >
> > > We thank the reviewer taking the time to review and confirm our response.

---

### Official Review · Reviewer_vR2R · 2023-07-28

**Rating:** 7
**Confidence:** 3

**Strengths:**

- The proposed benchmark is useful for both the neuroscience community (provides novel insights for brain connectivity), as well as the graph learning community (a useful application of graph neural networks).
- The benchmark is very extensive, the authors test various graph models across multiple tasks.


**Additional Feedback:**

This benchmark is designed to maximize the predictive power of graph models on brains. How does maximizing predictive power, provide novel insights into connectivity in the brain?

**Clarity:**

The paper is well written. The authors do a great job at motivating the different stages of this benchmark.

**Correctness:**

The dataset is constructed in a sound way. The authors do a great job at establishing the research questions and then providing extensive experimental results that explore these questions.

**Documentation:**

Yes.

**Ethics:**

No.

**Limitations:**

There are no obvious limitations.

**Opportunities For Improvement:**

- In section 5.2, it is concluded that a sparser graph significantly enhances the performance of GNNs. This could actually be a limitation of message passing neural network, which can easily suffer from the over-smoothing phenomena, where nodes converge to their neighbors' representations very quickly, and this might be worsened in a densely connected graph. Have the authors explore the use of edge weights or edge attributes? Maybe other types of graph models that have a gating mechanism to limit the flow of information in the graph and reduce oversmoothing?

**Relation To Prior Work:**

The authors clearly discuss previous work.

**Summary And Contributions:**

This paper introduces a novel collection of datasets consisting of both static and dynamic graphs extracted from fMRI data. A graph representation of this type of data appears to be natural, and can potentially capture and model the interactions between different brain areas.

---

> ### Author Response · Authors · 2023-08-22
> **Rebuttal by Authors**
>
> We greatly appreciate the insightful feedback provided by the reviewer as well as acknowledging the significance of our work. In the following, we respond to their comments.
>
> **Question 1:** In section 5.2, it is concluded that a sparser graph significantly enhances the performance of GNNs. This could actually be a limitation of message passing neural network, which can easily suffer from the over-smoothing phenomena, where nodes converge to their neighbors' representations very quickly, and this might be worsened in a densely connected graph. Have the authors explore the use of edge weights or edge attributes? Maybe other types of graph models that have a gating mechanism to limit the flow of information in the graph and reduce oversmoothing?
>
> **Response:** We agree with the reviewer that the density of the graph might potentially be causing oversmoothing, which can negatively impact the models' performance. Our results suggest that sparser and larger graphs often lead to improved results. However, the results are mixed. In some cases, such as those presented in Table 3, $k-$GNN, ResGCN, GIN, GAT, and ChebConv exhibit the best performance on dense graphs. However, the ratio of sparse to medium to dense graphs in the gender classification dataset is 8:0:2, while in the activity dataset, it stands at 4:2:4 as indicated in Table 3. Therefore, we think that the observed effect might indeed be attributed to oversmoothing, but further investigations are necessary to confirm its presence. We chose the sparse setting due to its dual advantages: it produces better results in the majority of cases and also has low computational and memory complexity.
> As for the incorporation of edge weights and edge attributes, we haven't delved into this facet; however, we believe that including such weights/attributes—likely reflecting correlations among ROIs could potentially open up another avenue for research exploration. While we have not employed specialized GNN convolutions to mitigate the over-smoothing effect, we have used Transformer convolution (UniMP) and Graph Attention Networks (GAT), both of which incorporate attention mechanisms. Notably, we haven’t observed any major difference between their results. Moreover, we appreciate the reviewer highlighting this interesting research direction for our benchmarks. We hope that this benchmark promotes further investigations into these questions.
>
> **Question 2:** This benchmark is designed to maximize the predictive power of graph models on brains. How does maximizing predictive power, provide novel insights into connectivity in the brain?
>
> **Response:** We thank the reviewer for highlighting this interesting question. We believe that models exhibiting greater predictive capabilities have learned meaningful insights regarding brain function, structure, and pathology. Such insights can illuminate the functional significance of specific brain regions in relation to the prediction task. For instance, graph models can identify pivotal nodes within the brain that exert a significant influence on overall connectivity and predictive efficacy [1]. These influential nodes may correspond to regions pivotal for information integration, and the corresponding prediction efficacy. Owing to space constraints, we were only able to include the visualizations of embeddings in Supplementary Material, section D, which has also been historically used for providing insights into high-level properties of networks. Interpreting these models poses challenges; nevertheless, this facilitates the execution of experiments involving ablation or alternative testing methods, aiming to discern differences in model behavior in relation to the network topology.
>
>
>  [1].Ying, Z., Bourgeois, D., You, J., Zitnik, M., \& Leskovec, J. (2019). Gnnexplainer: Generating explanations for graph neural networks. Advances in neural information processing systems, 32.

---

### Decision · Program_Chairs · 2023-09-22

**Decision:**

Accept (Poster)

**Comment:**

This submission generated much discussion. The reviewer appreciated the work that contributes an extensive benchmark of Graph Neural Networks for fMRI connectomics.

One suggestion that I (area chair) would have is to add non GNN baselines to the benchmark. Indeed, in a blinded challenge, Traut 2022 [1] found that GNNs underperformed simpler approaches.

[1] "Insights from an autism imaging biomarker challenge: Promises and threats to biomarker discovery", Traut et al 2022, NeuroImage